# Brain Bandit: A Biologically Grounded Neural Network for Efficient Control of Exploration

**Chen Jiang**[1]*, **Jiahui An**[2,1], **Yating Liu**[3,2,1], **Ni ji**[2,1]†
[1]Chinese Institute for Brain Research, Beijing
[2]Chinese Academy of Medical Sciences & Peking Union Medical College
[3]China Agricultural University
chen.jiang3@mail.mcgill.ca,{anjiahui, liuyating, niji}@cibr.ac.cn

## Abstract

How to balance between exploration and exploitation in an uncertain environment is a central challenge in reinforcement learning. In contrast, humans and animals have demonstrated superior exploration efficiency in novel environments. To understand how the brain's neural network controls exploration under uncertainty, we analyzed the dynamical systems model of a biological neural network that controls explore-exploit decisions during foraging. Mathematically, this model (named the Brain Bandit Net, or BBN) is a special type of stochastic continuous Hopfield network. We show through theory and simulation that BBN can perform posterior sampling of action values with a tunable bias towards or against uncertain options. We then demonstrate that, in multi-armed bandit (MAB) tasks, BBN can generate probabilistic choice behavior with a flexible uncertainty bias resembling human and animal choice patterns. In addition to its high efficiency in MAB tasks, BBN can also be embedded with reinforcement learning algorithms to accelerate learning in MDP tasks. Altogether, our findings reveal the theoretical foundation for efficient exploration in biological neural networks and propose a general, brain-inspired algorithm for enhancing exploration in RL. The code is available at https://github.com/Chen-Ginger/BrainBandit

## 1 Introduction

The explore-exploit (E-E) dilemma, originally described in the context of animal foraging (Stephens & Krebs, 1986; Charnov, 1976), has become an important problem across many fields including psychology, neuroscience and reinforcement learning (RL)(Addicott et al., 2017). Despite the development of numerous algorithms, sample-efficient exploration in RL remains difficult for complex, sparse-reward tasks (Sutton & Barto, 2018). Meanwhile, studies in humans and animals have revealed a diverse array of exploration strategies (Wilson et al., 2021; Schulz & Gershman, 2019). In addition, excitingly, recent research has begun to reveal the biological neural networks that give rise to the rich and flexible exploration behaviors (Costa et al., 2019; Tomov et al., 2020; Hogeveen et al., 2022; Costa & Averbeck, 2020). Based on recent findings in the biological neural network that controls exploration, we built the Brain Bandit Network (BBN), a stochastic Hopfield network for controlling exploratory action selection under input uncertainty. We show theoretically that the BBN model can perform Bayesian posterior sampling while implementing a tunable bias that ranges from optimistic, neutral, and conservative in the face of uncertainty.

Our main contributions are four-fold:

1. We propose a biologically grounded, scalable network model for solving the E-E dilemma.
2. We analytically show that BBN implements a hybrid between Bayesian posterior sampling and uncertainty-directed exploration.

---

*Current address: McGill University.
†Corresponding author.

3. We show that BBN can closely approximate human and animal behavior in bandit tasks under a variety of conditions.

4. We show that BBN can drive highly efficient exploration in bandit and MDP tasks, promising further application to more complex RL problems.

## 2 BACKGROUND AND RELATED WORK

### 2.1 THE EXPLORATION PROBLEM IN REINFORCEMENT LEARNING

The domain of efficient exploration in reinforcement learning focuses on balancing immediate rewards (exploitation) and information gathering for future rewards (exploration). A classic example is the Multi-Armed Bandit (MAB) problem, introduced by (Robbins, 1952) in 1952 and widely used to model this tradeoff (Lai & Robbins, 1985; Berry & Fristedt, 1985; Agrawal, 1995; Auer et al., 1995; Sutton & Barto, 1999). Conventional methods inject noise into action selection (Sutton & Barto, 1999), but these dithering algorithms can be inefficient. Alternative methods like Upper Confidence Bound (UCB) employ optimism in the face of uncertainty (OFU) by biasing for uncertain choices (Lai & Robbins, 1985; Agrawal, 1995; Auer et al., 1995). Thompson sampling (Thompson, 1933) makes decisions based on posterior samples rather than optimistic estimates. Optimistic Thompson Sampling (O-TS), combining UCB and Thompson sampling, reshapes the posterior distribution optimistically and exhibits strong empirical and theoretical performance (Chapelle & Li, 2011; May et al., 2012). More recent methods leverage deep networks to learn the exploration bonus (Zhou et al., 2020; Ban et al., 2022) or the posterior variance (Zhang et al., 2021).

### 2.2 BIOLOGICAL SOLUTIONS TO THE EXPLORE-EXPLOIT DILEMMA

Early work on the explore-exploit tradeoff, rooted in Optimal Foraging Theory and the Marginal Value Theorem (Stephens & Krebs, 1986; Charnov, 1976), suggests that animals achieve near-optimal balance between exploiting known resources and exploring uncertain options. Cognitive scientists have used bandit tasks to study this tradeoff in humans and animals (Addicott et al., 2017; Cohen et al., 2007; Wang et al., 2023; Beron et al., 2022). Two main strategies emerge: random exploration, involving stochastic action choices, and directed exploration, leveraging uncertainty to guide actions (Wilson et al., 2021; Schulz & Gershman, 2019). Humans and animals often combine these strategies flexibly, adjusting based on task horizon, option novelty, developmental stage, and mental state (Gershman, 2018; Bartumeus et al., 2016; Wilson et al., 2014; Cockburn et al., 2022; Mizell et al., 2024; Schulz et al., 2019; Addicott et al., 2017; Fan et al., 2023; Waltz et al., 2020). Additionally, they exhibit persistent exploration, repeating previous choices regardless of value (Beron et al., 2022; Laurie et al., 2024). These strategies resemble algorithms like Thompson sampling and Optimism in the Face of Uncertainty (OFU), but with key differences (Wilson et al., 2021).

To understand the brain's solution to the E-E dilemma, neuroscientists have identified neurobiological mechanisms that control explore-exploit decisions (Daw et al., 2006; Costa et al., 2019; Hogeveen et al., 2022). Recent studies in *C. elegans* (Flavell et al., 2013; Ji et al., 2021) have revealed a compact recurrent network governing the transitions between behavioral states analogous to exploration and exploitation (Fig. 1). This minimal network provides a unique opportunity to explore the algorithmic principles the brain uses to solve the E-E problem.

## 3 MODEL

### 3.1 THE BRAIN-INSPIRED BANDIT NETWORK (BBN) IS A STOCHASTIC CONTINUOUS HOPFIELD NETWORK

To model the biological neural network that controls E-E decisions during foraging (Fig. 8 (Ji et al., 2021)), we define a set of $N$ neurons whose temporal dynamics are described by the following stochastic differential equations (or Langevin equations):

$$\tau_i \frac{dx_i}{dt} = -\gamma_i x_i + \sum_{j \neq i}^{N} w_{ij} f(x_j) + b_i + \bar{I}_i + \sigma_i dW(t) \tag{1}$$

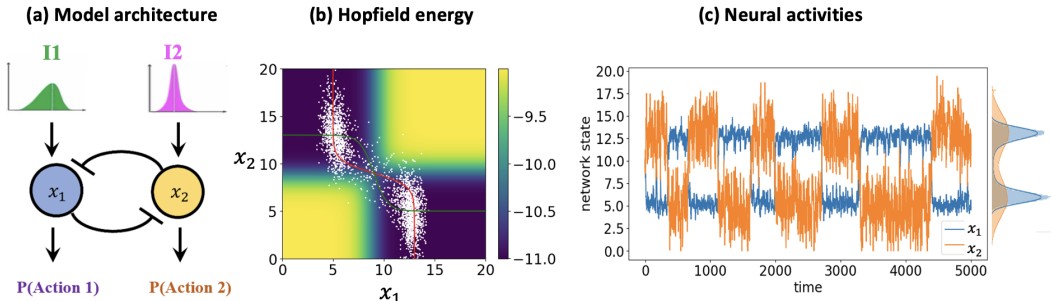

Figure 1: **The Brain-inspired Bandit Network (BBN)** (a) Architecture of the 2-D BBN model. (b) Hopfield energy (or Lyapunov function) plotted over the state space of BBN. Heatmap represents the Hopfield energy; red and green curves are the nullclines, and white dots represent simulated network states. (c) Evolution of network states over time

Where $f(x) = \frac{1}{1+e^{-n(x-k)}}$, $w_{i,j} < 0$, and $dW(t)$ is the Wiener process. Here, $w_{ij}f(x_j)$ represents the inhibitory interaction between neurons; $b_i$ is the baseline activity of neuron $i$; $\overline{I_i}$ and $\sigma_i dW(t)$ are the deterministic and the stochastic components of the external input, respectively. $\sigma_i$ is the standard deviation of the Wiener noise. We term this type of stochastic continuous Hopfield network with all negative weights the Brain-inspired Bandit Network (BBN), for reasons that will become clear later.

Assuming approximately symmetric weights $i.e., w_{ij} = w_{ji}$ [1], the deterministic part of the model is essentially a continuous Hopfield network (Hopfield, 1982; 1984) with exclusively inhibitory connections. It is hence associated with a Hopfield energy or Lyapunov function of the form:

$$E = \left\{ -\frac{1}{2} \sum_{i,j,i\neq j}^{N} w_{ij} f(x_i) f(x_j) + \sum_{i}^{N} \left[ x_i f(x_i) - \int_0^{x_i} f(x)dx \right] - \sum_{i}^{N} b_i f(x_i) \right\}$$

$$- \left\{ \sum_{i}^{N} \bar{I}_i f(x_i) \right\} = E^{int} - E^{ext} \tag{2}$$

Here, we have decomposed the Hopfield energy $E$ into $E^{int}$, dependent only on internal network parameters, and $E^{ext}$, which embodies influence from external inputs $\overline{I_i}$. With suitable parameters (see Appendix B.1), the model can have up to $N$ local energy minima or attractor states exhibiting winner-take-all dynamics (Fig. 1 and Fig. 12). Stochastic noise induces transitions between these attractor states, consistent with experimental findings in foraging networks (Ji et al., 2021).

## 3.2 THE BBN IMPLEMENTS BAYESIAN POSTERIOR SAMPLING

Hinton and Sejnowski (Hinton & Sejnowski, 1983) have demonstrated that a discrete Hopfield network with stochastically activating units (i.e. an Ising network) can implement Bayesian inference by sampling from the posterior distribution. Here we extend this conclusion to continuous Hopfield networks. Briefly, using Kramers' escape theory (Kramers, 1940; Langer, 1968; Hänggi et al., 1990), we can approximately compute the mean first passage time (MFPT), defined here as the expected time to leave an attractor state $\mathcal{A}$ and crossing the nearby saddle point $\mathcal{S}$, as:

$$\langle \tau_A \rangle = \frac{2\pi\gamma}{\omega_b} \frac{\prod_i' \omega_i^S}{\prod_i \omega_i^A} * \exp\left( \frac{\Delta E_A}{D_A} \right) \tag{3}$$

---

[1]While the original Hopfield network study (Hopfield, 1984) required weight symmetry to prove absolute stability of the energy (or Lyapunov) function. Later work (Matsuoka, 1992; Chen & Amari, 2001) have shown that the global convergence of the Hopfield energy function still holds for networks with asymmetric weights.

Where $\gamma$ is the friction coefficient (equivalent to $\tau$ in Eq. 1, and $\omega_i^A$ are the angular frequencies (i.e. eigenvalues of the Hessian matrix) at the center (i.e. energy minimum) of the attractor. $\omega_b$ and $\omega_i^S$ are the angular frequencies of the saddle point, with $\omega_b$ associated specifically with the unstable mode. $\Delta E_A$ is the energy difference between the saddle point and the center of the attractor and $\Delta E_{A \to S} = E_S - E_A$. $D_A$ is the diffusion constant, which in thermodynamics scales with the magnitude of the stochastic noise.

The equilibrium probability of the network being in a given attractor state $A_1$ can be approximated by its stability, measured via the MFPT, relative to the other attractors. This translates to:

$$P_{A_1} \cong \frac{\langle \tau_{A1} \rangle}{\sum_1^N \langle \tau_{Aj} \rangle} = \frac{1}{1 + \sum_2^N \left\{ \frac{\alpha_j}{\alpha_1} \exp \left( \frac{\Delta E_{Aj}}{D_{Aj}} - \frac{\Delta E_{A1}}{D_{A1}} \right) \right\}}, \quad \text{where } \alpha_{i \in \{1,\ldots,N\}} = \frac{\prod_j' \omega_j^{S_i}}{\omega_b \prod_j \omega_j^{A_i}}$$

(4)

Assuming identical biophysical parameters and inputs for all neurons, the angular frequencies $\omega_j^{A_i}$ of the $N$ attractors are permutations of each other and there is a single saddle point defined by $x = \frac{1}{\gamma} N w f(x) + b + \bar{I}$. This leads to $\alpha_1 = \alpha_j, \forall i$. Further, by substituting $\Delta E_A = (E_S - E_A^{int}) + E_A^{ext}$ into Eq. 4, we have:

$$P_{A_1} \cong \frac{1}{1 + \sum_2^N \left\{ \exp \left( \left[ \frac{E_S - E_{Aj}^{\text{int}}}{D_{Aj}} - \frac{E_S - E_{A1}^{\text{int}}}{D_{A1}} \right] + \left[ \frac{E_{Aj}^{\text{ext}}}{D_{Aj}} - \frac{E_{A1}^{\text{ext}}}{D_{A1}} \right] \right) \right\}}$$

(5)

Now if we define the probability of an attractor state in the absence of external inputs as its prior probability as: $P_{Ai}^{\text{prior}} = \exp \left( \Delta E_{Ai}^{\text{int}} / D_{Ai} \right)$, and the probability of the state given input data (e.g. sensory evidence) as: $P \left( \bar{I} \mid A_i \right) = \exp \left( E_{Ai}^{\text{ext}} / D_{Ai} \right)$, we have:

$$P \left( A_1 \mid \boldsymbol{I} \right) \cong \frac{1}{1 + \sum_2^N \left\{ \left( P_{Aj}^{\text{prior}} / P_{A1}^{\text{prior}} \right) * \left[ P \left( \boldsymbol{I} \mid A_j \right) / P \left( \boldsymbol{I} \mid A_1 \right) \right] \right\}}$$

(6)

Eq. 6 reveals a close connection between the Hopfield-energy-based formulation of attractor state probability and Bayesian inference. Specifically, if we consider $P_{Ai}$ as the probability of a hypothesis $i$ being true or a decision $i$ being optimal, then Eq. 6 essentially computes the Bayesian posterior of $i$ given external evidence.

## 3.3 The BBN can exhibit *OPTIMISTIC*, *NEUTRAL*, or *CONSERVATIVE* biases on input uncertainty

In Kramers' theory, the diffusion constant $D$ from thermal fluctuations is typically isotropic ($\boldsymbol{\Sigma} = \sigma^2 I$, $D = \sigma^2$). However, in our model, input to each neuron can have different levels of uncertainty, making the overall noise anisotropic. Recent studies (Zhu et al., 2018; Yang et al., 2023) show that anisotropic noise affects escape efficiency (the rate at which model leaves one of its attractor states (i.e. 1/MFPT)) by interacting with local attractor curvature. Starting from a local energy minimum at $x_0$, the model evolves as:

$$\langle E \left( x_t \right) \rangle \cong E \left( x_0 \right) - \int_0^t \left\langle \nabla E^T \nabla E \right\rangle + \frac{t}{2} \left\langle \text{Tr} \left( \boldsymbol{H_0} \boldsymbol{\Sigma} \right) \right\rangle$$

(7)

Here, $\boldsymbol{H_0}$ is the Hessian matrix evaluated at the attractor bottom, and $\boldsymbol{\Sigma}$ is the noise covariance matrix. Since both matrices are diagonal in our model, the escape efficiency is highest when the dimensions of largest input noise align with those of highest curvature. To capture this effect, we define an isotropic noise $\overline{\boldsymbol{\Sigma}} = \overline{\sigma}^2 I$ that yields the same efficiency as $\boldsymbol{\Sigma}$:

$$\text{Tr}(\boldsymbol{H_i} \overline{\boldsymbol{\Sigma}}) = 2\overline{\sigma_i}^2 \, \text{Tr}(\boldsymbol{H_i}) = \text{Tr}(\boldsymbol{H_i} \boldsymbol{\Sigma}), \quad \text{where } \overline{\sigma_i}^2 = \frac{\text{Tr}(\boldsymbol{H_i} \boldsymbol{\Sigma})}{\text{Tr}(\boldsymbol{H_i})} = D_i^{\text{eff}}$$

(8)

Here, $D_i^{\text{eff}}$ represents the effective diffusion constant and $\boldsymbol{H_i} = \boldsymbol{PH_j} = \boldsymbol{H_A}, \forall i$ where $\boldsymbol{P}$ is a permutation matrix. Substituting Eq. 8 into Eq. 4 , we have:

$$P_{A1} = \frac{1}{1 + \exp\left\{2\operatorname{Tr}(\boldsymbol{H_A})\Delta E_A \left(\frac{1}{\operatorname{Tr}(\boldsymbol{H_A^T\Sigma})} - \frac{1}{\operatorname{Tr}(\boldsymbol{H_A\Sigma})}\right)\right\}} \tag{9}$$

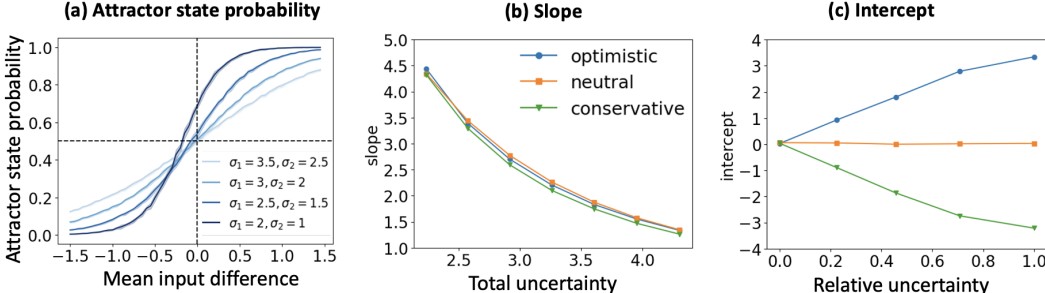

Figure 2: **BBN implements Bayesian posterior sampling with a tunable bias towards, neutral to, or against input uncertainty.** (a) Sigmoidal dependence of attractor state probability on the difference in mean input values. (b) Slope of the state probability curve in (a) as a function of total input uncertainty (defined as $\sqrt{\sigma_1^2 + \sigma_2^2}$) for the three types of networks. (c) Intercept of the state probability curve as a function of relative input uncertainty (defined as $\sigma_1 - \sigma_2$).

While all $N$ attractors have equal energy and share a common set of angular frequencies, their Hessian matrices are non-identical and can interact differently with non-isotropic noise (i.e.,$\boldsymbol{\Sigma} \neq c\boldsymbol{I}$). If $P_{A1}$ corresponds to the attractor state with the highest input noise, the following scenarios can occur (assuming $j \neq 1$):

1. $\operatorname{Tr}(\boldsymbol{H_1\Sigma}) < \operatorname{Tr}(\boldsymbol{H_j\Sigma})$ and $P_{A1} > P_{Aj}$ (**Optimistic**). 2. $\operatorname{Tr}(\boldsymbol{H_1\Sigma}) = \operatorname{Tr}(\boldsymbol{H_j\Sigma})$ and $P_{A1} = P_{Aj}$ (**Neutral**). 3. $\operatorname{Tr}(\boldsymbol{H_1\Sigma}) > \operatorname{Tr}(\boldsymbol{H_j\Sigma})$ and $P_{A1} < P_{Aj}$ (**Conservative**).

These regimes are termed as **Optimistic**, **Neutral**, and **Conservative**, respectively. Fig. 2 illustrates the input dependence of attractor state probabilities under these three regimes.

Parameter sensitivity analyses (Fig. 3(a-b) and Fig. 11 in Appendix A) reveal that the three parameter regimes span a wide range of parameter combinations, obviating the need for fine-tuning. By adjusting the baseline activity $b$, synaptic threshold $k$, or inhibitory synaptic weight $w$ — either individually or in pairs — one can flexibly modulate the uncertainty bias from highly optimistic ($P_{A1} \to 1$) to neutral ($P_{A1} = \frac{1}{N}$) to highly conservative ($P_{A1} \to 0$).

### 3.4 (OPTIMISTIC) UNCERTAINTY BIAS IS PRESERVED IN HIGHER DIMENSIONS

The theoretical analysis above predicts that the uncertainty bias of BBN should scale well to high dimensions. To verify this empirically, we progressively increased network dimension by adding more neurons, while keeping all network parameters in Eq. 1 unchanged. Strikingly, for a BBN that is optimistic at $N = 2$, scaling up to $N = 10$ did not alter its optimistic bias (Fig. 3(c)). In contrast, a BBN that is neutral in 2D became mildly optimistic as $N$ increased, while a conservative BBN becomes mildly optimistic at $N > 5$. Thus, with increasing network dimension, the model exhibits a tendency to bias towards attractor states with higher input uncertainty.

To understand this empirical phenomenon, we examined state-transition dynamics near the saddle point for a perfectly neutral 3D BBN (i.e., $\boldsymbol{H_i} = c\boldsymbol{I}, \forall i$) (Fig. 13). With isotropic noise, the network exhibited equal probability of entering any attractor state. However, with highly anisotropic noise, it preferentially entered the attractor state along the dimension of highest noise, creating a bias towards high-uncertainty states. This makes conservative bias harder to maintain and optimistic bias more prominent in high-dimensional models (Fig. 14. To incorporate this effect into our theoretical framework, we need to combine escape rates analysis (Kramers, 1940; Zhu et al., 2018) with theory

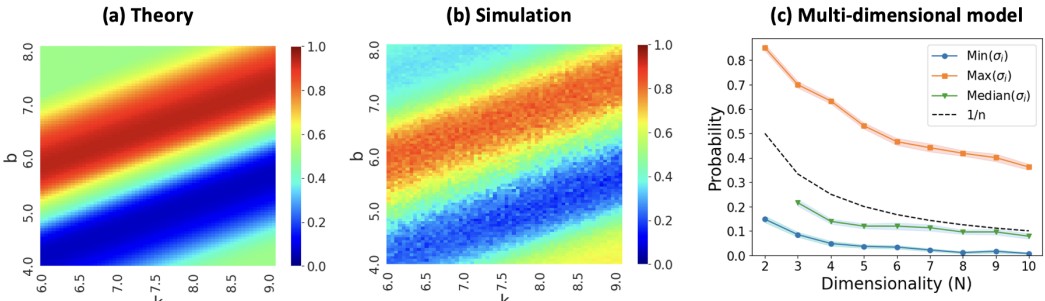

Figure 3: **Parameter dependence and multi-dimensional model.** (a) Theoretically derived and (b) numerically simulated attractor state probability as a function of network parameters $b$ and $k$. The color scale corresponds to the probability that the network samples the attractor state driven by the highest input uncertainty, which is an indicator of the network's uncertainty bias. (c) Equilibrium attractor state probabilities in high dimensional BBN models. Three colored lines correspond to attractor states driven by the highest (orange), median (green), or lowest (blue) levels of input uncertainty. The network parameters remain unchanged as the dimensionality N increases.

of dynamics around saddle points (Daneshmand et al., 2018)—a challenge we aim to address in future work.

# 4 EXPERIMENTAL EVALUATION

## 4.1 UNCERTAINTY-AWARE EXPLORATION IN MULTI-ARMED BANDIT TASK

Given BBN's ability to infer and sample from a posterior distribution with a tunable uncertainty bias, a natural application of BBN is to control action choice given external, uncertain evidence. We thus adapted the BBN model to play multi-armed bandit (MAB) games and compared its performance with classic bandit algorithms.

### 4.1.1 RUNNING BBN IN BANDIT GAMES

To make the BBN model play bandit games, we (1) define a BBN model with $N$ neurons, each corresponding to one of the $N$ bandit arms; (2) pick network parameters that yield "optimistic" exploration for a 2-D BBN, and simply apply the same parameters to all neurons in the N-D model; (3) prior to each bandit trial, assign network input $\boldsymbol{I}$ by sampling from the reward memory buffer and numerically simulate the network for $T$ steps using the Runge-Kutta method; (4) at the end of simulation, select the arm $a$ whose corresponding neuron has the highest activation value; (5) collect the reward $r_a$ and add it to the memory buffer for arm $a$; (6) repeat (3)-(5) for the next trial till the game ends. The pseudocode along with detailed task parameters are presented in Appendix B.1.

### 4.1.2 BBN IMPLEMENTS UNCERTAINTY-AWARE POSTERIOR SAMPLING

To reveal BBN's exploration strategies, we examined the dependence of choice probability on total and relative reward uncertainty for BBN agents with optimistic, neutral, or conservative biases, as well as classic algorithms Thompson Sampling (TS) and Upper Confidence Bound (UCB). As shown in Fig. 4 (a-b), TS exhibited a constant intercept regardless of relative uncertainty (RU) and a decreasing slope with increasing total uncertainty (TU), indicating sensitivity only to total uncertainty; UCB exhibited a constant slope with varying TU and an increasing intercept with increasing RU, indicating sensitivity only to relative uncertainty. In contrast, BBN with optimistic parameters showed variation in both slope and intercept with changes in TU and RU. These results indicate that BBN implements a hybrid algorithm combining posterior sampling (like TS) with tunable bias towards high uncertainty (akin to UCB).

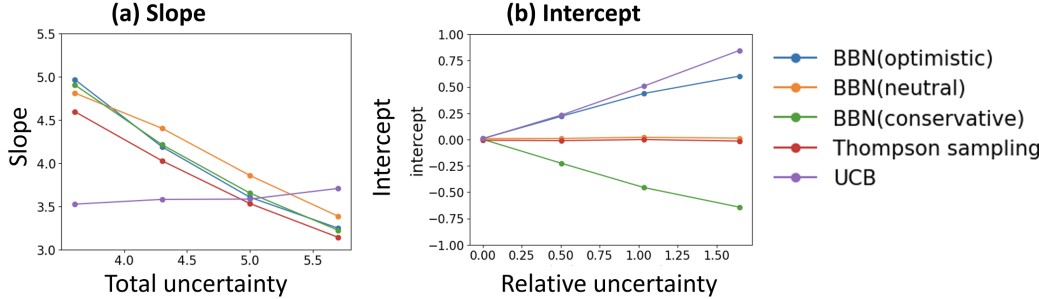

Figure 4: **Exploratory behavior of BBN, Thompson sampling and UCB in 2-armed bandit games** (a) Slope of the choice probability curve as a function of total uncertainty. (b) Intercept of the choice probability curve as a function of relative uncertainty.

### 4.1.3 EFFICIENT EXPLORATION IN BANDIT TASKS

We next compared the empirical performance of BBN-driven exploration in comparison against UCB, Thompson sampling, and Optimistic Thompson Sampling (OTS, (Hu et al., 2023)) in both 2-armed bandit and 3-armed bandit games. Each agent played 10,000 game blocks of 20 trials each in 2-armed bandit games and 30 trials each in 3-armed bandit games. Fig. 5 (a-b) presents the probability of choosing the optimal arm as trial number increases. BBN (with optimistic parameters) consistently outperformed other algorithms in 2-armed bandits and topped the performance in 3-armed bandit games. The other 'hybrid' algorithm, OTS, performed close to BBN in 3-armed bandits, but did poorly in 2-armed bandits.

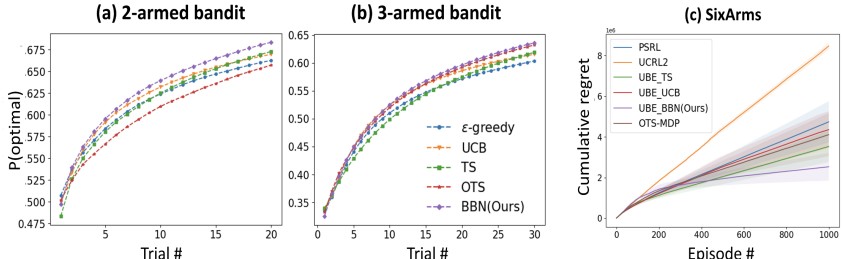

Figure 5: **BBN achieves efficient exploration in both bandit tasks.** (a) The probability of choosing optimal action over trials in 2-armed bandit games. (b) The probability of choosing optimal action over trials in 3-armed bandit games. (c) Cumulative regret in the SixArms (MDP task, see Fig. 16)

### 4.2 BBN CLOSELY APPROXIMATES BANDIT CHOICE BEHAVIOR IN HUMANS AND ANIMALS

The results above indicate that BBN exhibits similar hybrid strategies as previously reported in humans (Wilson et al., 2014; Gershman, 2018). We thus asked whether BBN can accurately model human and animal choice patterns in bandit tasks. We first compiled several publicly available datasets of humans playing bandit games (detailed list in Appendix C). We performed optimization on two network parameters $b$ and $k$ to minimize the difference between the choice probability curves output by BBN and in the human datasets. As shown in Fig. 6 (a-b), BBN can closely fit to both the intercept and the slope of human choice probability curves. In contrast, Thompson sampling failed to fit to the diverse intercepts across human groups, and UCB consistently yielded slopes that are much higher than those observed in human data.

We next extended the above analyses to a dataset in which mice played switching blocks of 2-armed bandit games (Beron et al., 2022). In this dataset, the reward for each arm is sampled from a Bernoulli distribution. In addition, the mean reward for each arm is not static, with a small probability (0.02) of being reversed before each trial starts. Based on results from (Beron et al., 2022),

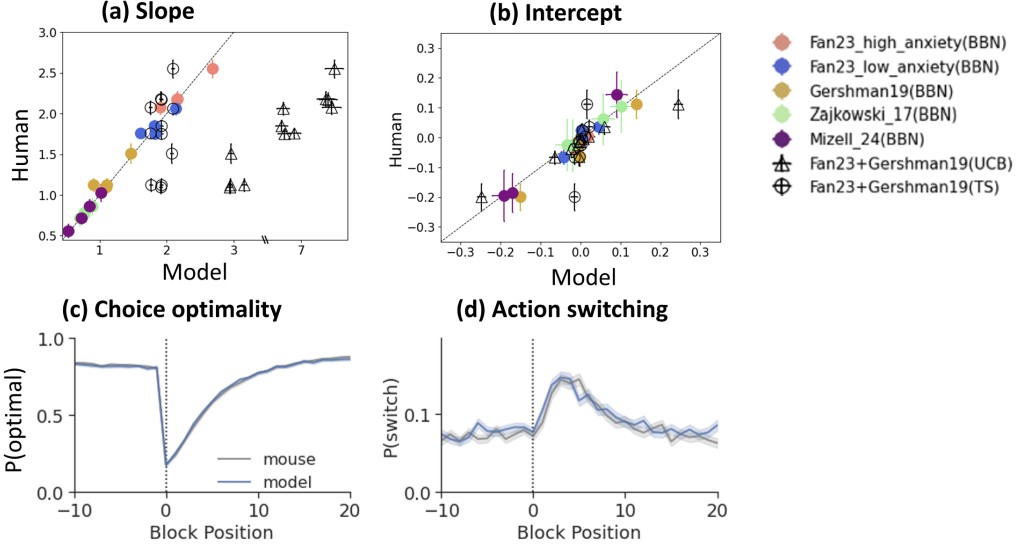

Figure 6: **The choice pattern of BBN closely approximates humans and animals in MAB tasks.**
(a-b) BBN-fitted versus actual slope and intercept values extracted from human data. (c-d) The
probability of choosing the optimal arm and switching to another arm upon block transition in mice
playing the 2-armed bandit game.

we used the last five rewards as inputs to the BBN model to drive choice behavior. As shown in Fig.
6 (c-d), parameter-tuned BBN generated choice and switching behavior that closely approximated
those exhibited in the mice study.

### 4.3 EFFICIENT EXPLORATION IN MDP PROBLEMS

Building on the strong performance of BBN in MAB tasks, we next applied BBN to MDP problems.
Unlike bandit problems with immediate rewards and no state transitions, MDP tasks require sequen-
tial decision-making under delayed rewards and unknown transition probabilities (Bellman, 1966;
Bertsekas, 2012). Among existing methods, UCRL2 (Auer et al., 2008) extends OFU to MDPs,
while PSRL (Strens, 2000; Osband et al., 2013) generalizes posterior sampling to RL. Hybrid algo-
rithms like Optimistic Thompson Sampling (OTS) (Agrawal & Jia, 2017; Tiapkin et al., 2022; Hu
et al., 2023) aim to improve exploration efficiency but face challenges such as computational cost
and uncertainty estimation.

We consider a finite-horizon MDP with state space $S$, action space $A$, horizon $H$, rewards $r_{sa}^l$, and
transition probabilities $\mathbf{P}_{sa}$ conditioned on states $s$, actions $a$, and step $l$. The expected total return
at step $l$ under policy $\pi$ can be estimated iteratively using the Bellman equation:

$$Q_{sa}^{t+1} = \mu_{sa} + \sum_{s'a'} \pi_{s'a'} P_{sas'} Q_{s'a'}^t$$

where $\mu = \mathbb{E}(r)$ is the mean reward. Estimating uncertainty in Q-values remains an open issue in
RL. Donoghue et al. (O'Donoghue et al., 2018) proposed the Uncertainty Bellman Equation (UBE)
to provide an upper bound on the variance of Q-value posteriors. For tabular state space, this method
effectively propagates local variance estimates to global value uncertainty.

### 4.3.1 RUNNING BBN IN MDP TASKS

To apply BBN to drive action-selection in MDP tasks, we (1) define a BBN model with $N$ neu-
rons, each corresponding to one of the $N$ discrete actions, select network parameters that belong to
the "optimistic" regime for a 2D network; (2) initialize state-action values to i.i.d. Gaussian distri-
butions; (3) sample input values for each neuron from the distributions of state-action values and

perform numerical simulation of the BBN network for $T$ steps using the Runge-Kutta method; (4) at the end of the simulation, select action $a$ whose corresponding neuron has the highest activation value; (5) collect the reward $r_a$ and move to the next state ; (6) Repeat (3)-(5) till the episode ends; (7) Update the distribution of state-action values using the Uncertainty Bellman Equation (UBE) algorithm(O'Donoghue et al., 2018). (8) repeat (3)-(7) for the next episode till the game ends. We present the pseudo-code for the Algorithm 2 in Appendix B.4.

We first compared the exploration efficiency of the BBN-based algorithm (UBE_BBN) on the SixArms (Strehl & Littman, 2008) task, with additional implementation details presented in Appendix B.5. We compare our model to PSRL(Osband et al., 2013), UCRL2 (Auer et al., 2008) and OTS-MDP (Hu et al., 2023). We also specifically tested the role of BBN by replacing it with UCB (UBE_UCB) or Thompson sampling (UBE_TS). In PSRL, we maintain a Gaussian distribution for the rewards and a Dirichlet distribution for the transition probabilities. In the OTS-MDP and BBN models, we follow(Hu et al., 2023) and limit our uncertainty estimation to the reward $r$ for simplicity. As shown in Fig. 5 (c), the cumulative regret is lowest in UBE-BBN, which demonstrates the potential of BBN in promoting highly efficient exploration.

### 4.3.2 GRID WORLD

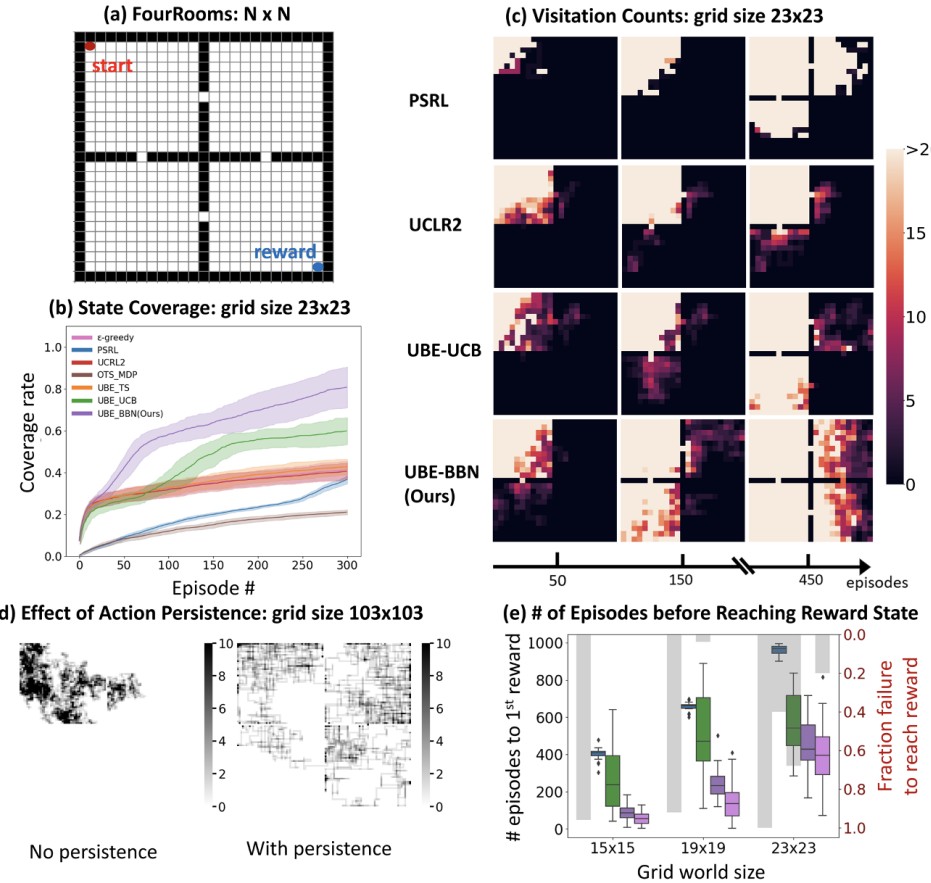

Figure 7: **BBN-enhanced RL agent exhibits efficient exploration in the FourRooms task.** (a) The FourRooms environment. The agent starts at the red point and can receive a reward only at the blue point. (b) The percent of grids covered (i.e. the coverage rate) by agents driven by various exploration algorithms over the period of training. (c) Display of visitation counts over the course of training. (d) Visitation counts for the UBE-BBN agent with or without action persistence. (e) Number of episodes taken till first reaching the reward state for different agents. Pink and purple are the UBE-BBN agents with and without action persistence respectively. Blue is PSRL and green is UBE_UCB

We next evaluated the exploration efficiency of BBN on sparse-reward MDP tasks, specifically the FourRooms task. In this task, an N-by-N grid world is divided into four compartments connected by narrow passages (Fig. 7 (a)). The agent starts from the upper left corner (red dot) and explores the environment to learn state-action values. First, we conducted reward-free exploration by assuming no rewards at any state. Exploration efficiency was measured as the coverage rate (ratio of visited states to total states) over episodes. Fig. 7 (b) shows that UBE-BBN achieved the fastest coverage rate among all methods. Fig. 7 (c) provides examples of cumulative visitation counts for each method during training. We then varied the environment size and repeated the experiments. UBE-BBN scaled well with grid size, while other algorithms faltered (Fig. 19 in Appendix E). Additional comparisons with more methods in different conditions are in Fig. 20-23 in Appendix E. Trajectories (visitation counts in a single episode) in Fig. 24 reveal that UBE-BBN excelled in extended deep exploration, covering hard-to-reach states effectively. Finally, we enhanced action persistence in UBE-BBN by allowing the BBN model to inherit activity states from the previous step (Fig. 25). This modification leveraged the Hopfield network's persistence property, instilling action correlation within episodes. As shown in Fig. 25, adding persistence further boosted UBE-BBN's exploration efficiency in the FourRooms task at large grid sizes.

**Parameter sensitivity in MDP tasks:** We additionally performed parameter sensitivity analysis for the SixArms and FourRooms task (as shown in Fig. 18 in Appendix E.1) and demonstrated that a broad range of "optimistic" network parameters yielded high performance on these tasks. Hence, optimistic BBN generally delivers good performance in these MDP tasks without requiring parameter fine-tuning.

## 5    DISCUSSION

We have demonstrated both theoretically and empirically that the BBN architecture can drive flexible and efficient exploration in ways similar to humans and animals. However, several limitations and open questions remain regarding its practical application. **First**, simulating the stochastic differential equations incurs high computational costs. This issue may be circumvented by analytically computing the attractor probabilities using Eq. 4 or by employing neuromorphic hardware. **Second**, given the development of many hybrid TS and OFU methods in the RL community (Hu et al., 2023; Tiapkin et al., 2022; Agrawal & Jia, 2017), it's intriguing to consider what gives rise to BBN's superior performance. One possibility is that BBN, as a system of coupled Langevin equations, effectively implements Langevin sampling of the posterior distribution. Langevin sampling has been shown to enjoy faster mixing and convergence rates than other sampling methods and is particularly well-suited for approximate Bayesian inference (Welling & Teh, 2011). **Third**, the current BBN algorithm lacks the ability to estimate uncertainty associated with state-action values, relying instead on a separate algorithm (in this case, the UBE) to generate value distributions. How biological neural networks compute and encode uncertainty remains an outstanding question, especially in sequential decision settings. Recent studies have suggested that a distributed population code (Dehaene et al., 2021) or a spatiotemporal activity pattern could encode uncertainty levels (Savin & Denève, 2014). We hope future experimental and theoretical studies will provide more insights into how the brain estimates and utilizes uncertainty. **Lastly**, given that humans and animals can flexibly modulate their uncertainty bias in a context-dependent manner, a valuable extension for the BBN algorithm would be to integrate contextual information into the network input. Expanding the BBN model to include upstream neurons found in the biological foraging network might help implement context-dependent E-E decisions (Fig. 8).

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

## APPENDIX

## A    SUPPLEMENTAL FIGURES

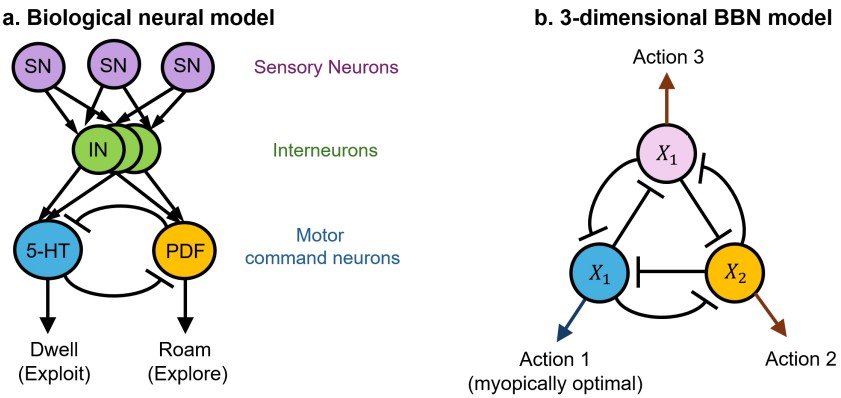

Figure 8: **(a)** A biological neural network in *C. elegans* that controls the exploration state (roaming) and exploitation state (dwelling). **(b)** Architecture of the 3-D BBN model.

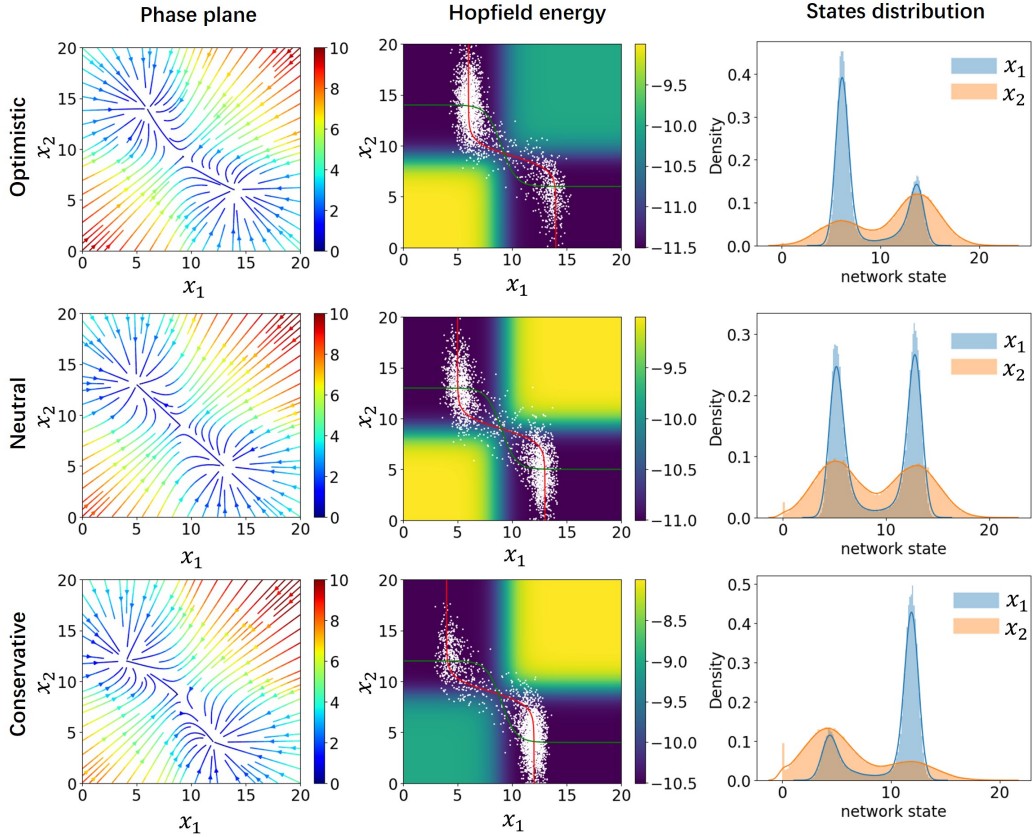

Figure 9: **Stability analysis on the three types of BBN models that generate *optimistic*, *neutral* or *conservative* bias to uncertainty.**

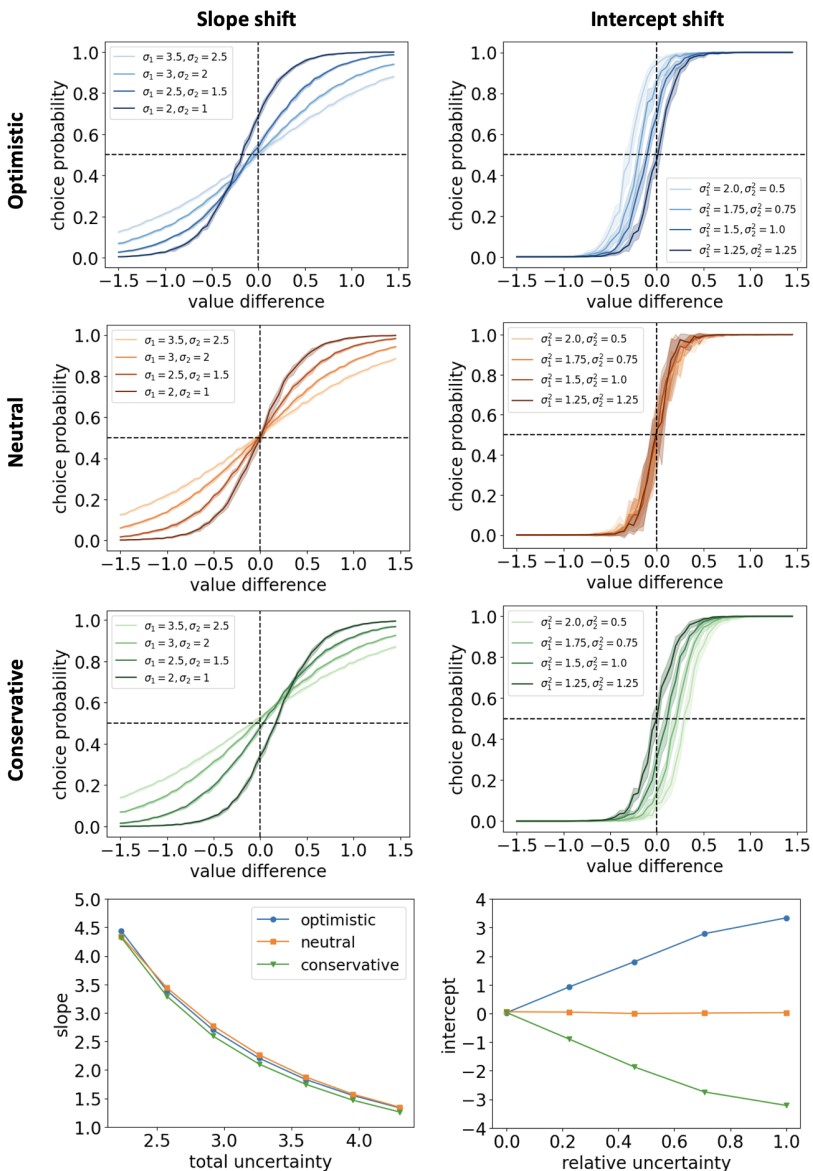

Figure 10: **Slope and intercept shift.** (Left column) The slope decreases as the total uncertainty increases while relative uncertainty is kept unchanged. (Right column) The intercept increases (**optimistic**), stays unchanged (**neutral**), or decreases (**conservative**) as relative uncertainty increases and total uncertainty kept unchanged.

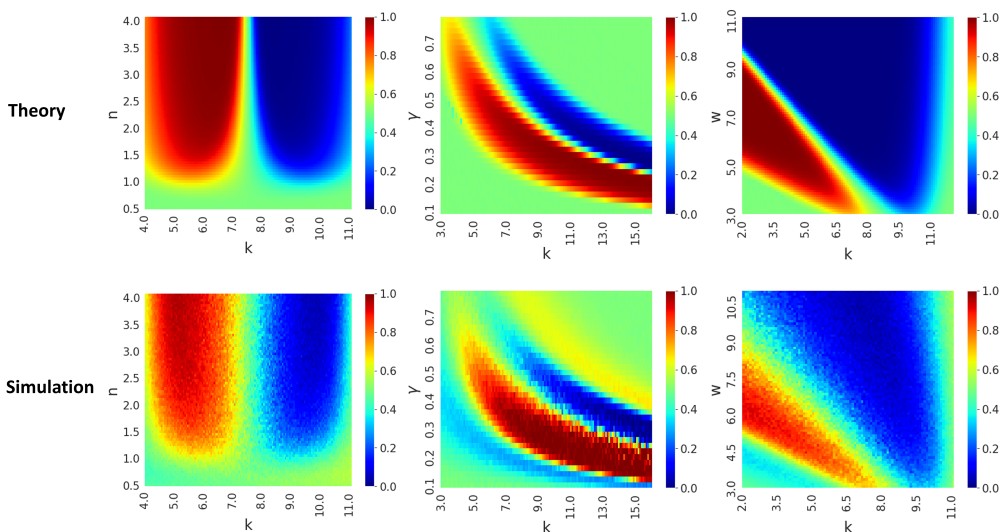

Figure 11: **Attractor state probability as a function of network parameters** Warm colors indicate a higher chance of finding the network in the state that receives greater input uncertainty.

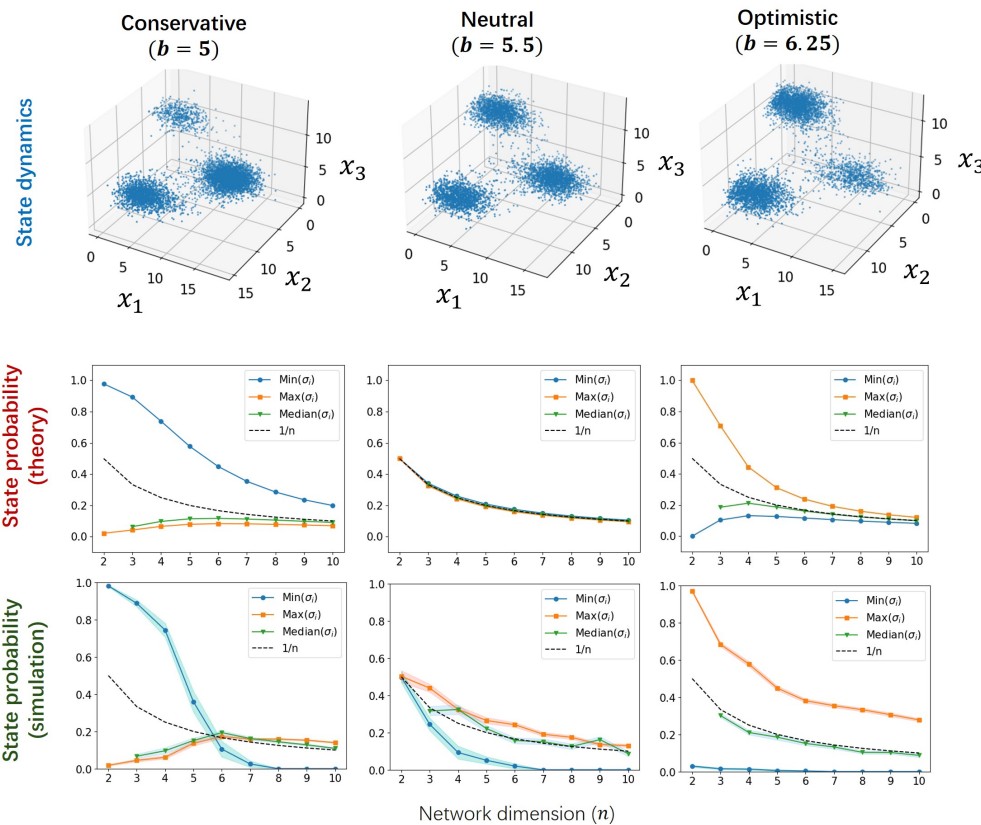

Figure 12: **Uncertainty bias in multi-dimensional BBN models.** (Top row) State dynamics of 3-D BBN model with conservative, neutral and optimistic uncertainty biases. The concentration of state dynamics reveal the three attractor states, which are visited with different relative proportion in the three types of BBNs. Input noise is strongest along the Z-direction is the largest and lowest along the X-direction. (Middle row) Probability of attractor states with the highest (orange), median (green), and lowest (blue) input uncertainty as the network scales from 2D to 10-D under the conservative, neutral, or optimistic parameter regimes, computed from numerical simulations. For the same type of network, internal model parameters are kept the same as the dimensionality increases. The dotted curve indicates perfectly equal partition of probability among all $N$ states. (Bottom row) Theoretically predicted state probability for the same network models presented above in the middle row, presented in the same format.

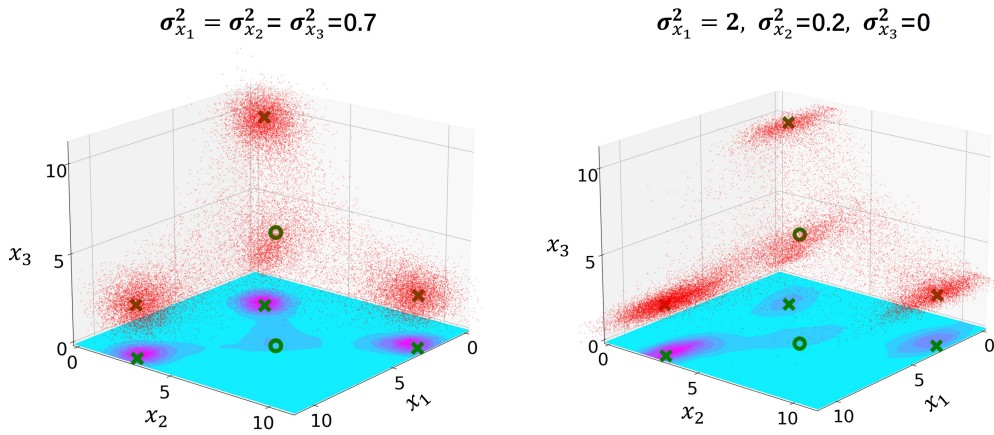

Figure 13: **State entry dynamics near the saddle point for a 3D BBN.** Red points show simulated state dynamics when the network is initialized from the saddle point. Green circle denotes the saddle point, green crosses denote the attractor centers, and the projected 2D histogram reveal the relative occupancy of the three attractors (pink indicates high state probability).

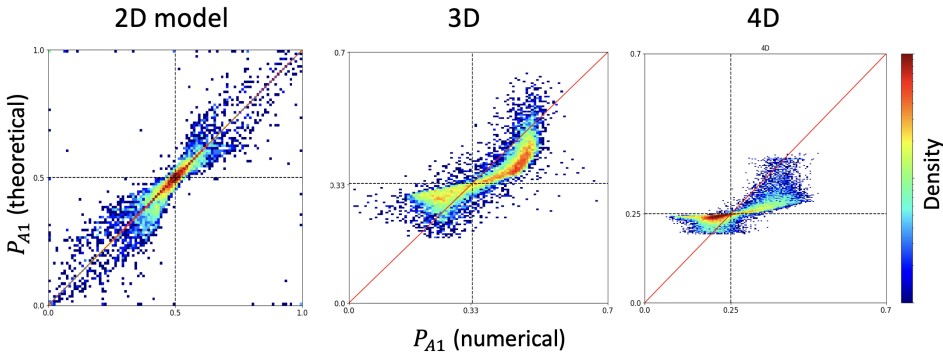

Figure 14: **Theoretical vs. simulated attractor state probability in multi-dimensional BBNs.**

## B  METHOD DETAILS

### B.1  PARAMETER SELECTION

| Parameter | Definition | Suggested range |
|---|---|---|
| $w$ | inhibitory weights | [2, 4], increase to make states more stable |
| $b$ | activity baseline | [5.5, 7], increase makes network optimistic |
| $k$ | threshold of sigmoid | [6, 8], increase makes network conservative |
| $n$ | slope of sigmoid | [1, 2], increase amplifies uncertainty bias |
| $\gamma$ | leak current or decay rate | 0.5 |
| $\tau$ | time constant | 1 |

Table 1: Internal Model Parameters

| Parameter | Definition | Suggested range |
|---|---|---|
| $I$ | mean of input | scale raw input values to [-2, 2] |
| $\sigma^2$ | variance of input | scale raw input variance to [0, 2] |

Table 2: External Parameters

| Parameter | Definition | Suggested range |
|:---:|:---:|:---:|
| $N$ | number of neurons | typically equal to number of actions choices |
| $T$ | total simulation steps | [400, 1000] |
| $dt$ | step length | 0.1 or 0.2 if using suggested parameter ranges |

Table 3: tab: Hyperparameters

In this section, we list the primary parameters used in BBN (Tables 1, 2, 3) and provide a principled way to determine optimal parameters for new environments.

Based on our experience and past literature (May et al., 2012; Hu et al., 2023; Agrawal & Jia, 2017), optimistic bias generally promotes efficient exploration. In addition, our sensitivity analysis on MDP tasks (Fig. 18) showed that a broad range of "optimistic" parameters yielded high performance, obviating the need for extensive fine-tuning. Further, we have shown that network parameters that yield optimistic bias for a 2D BBN preserve such bias in higher dimensions (Fig. 3(c) and Fig. 12). Thus, the steps to set up a $N$-dimensional BBN model are:

(1) Define a BBN model with $N$ interconnected neurons;

(2) Select internal network parameters 1 from the "optimistic" regime based on sensitivity analysis results presented in (Fig. 3(a-b) and Fig. 11), or use the parameter ranges suggested below as a starting point;

(3) Verify that the 2D network has two attractors and exhibits optimistic bias by numerically simulating the model under anisotropic 2D Gaussian noise with $\mu = [0, 0], \sigma = [1, 0.1]$); tune the parameters if necessary using the tips provided below;

(4) Apply these parameters to all neurons in the $N$D network;

(5) Scale the input to the network (typically past rewards or Q-values) to a range that permits the existence of multiple attractors (use suggested range or verify empirically).

We found that simulation step number of T=400 is sufficient for bandit and MDP tasks t. Below are sample network dynamics in the first episode of a 2-armed bandit game. Multiple transitions occurred between the attractor states, reflecting equal state probability as expected for equal uncertainty for the two arms.

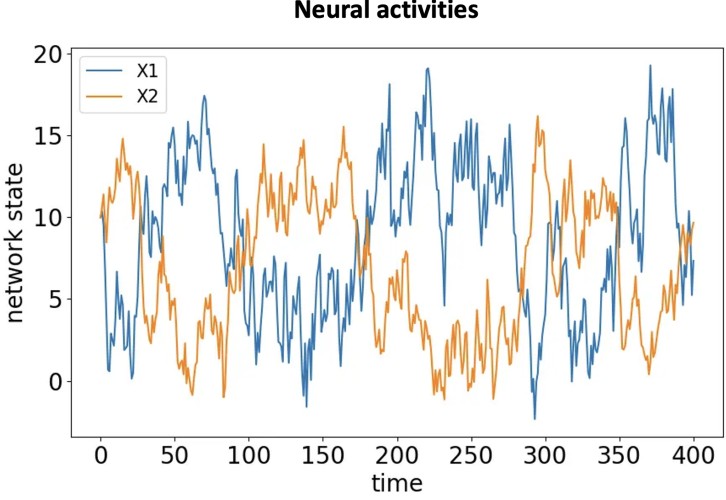

Figure 15: **State dynamics of BBN in a two-armed bandit game**

## B.2 Running BBN in bandit games

To make the BBN model play bandit games, we

(1) define a BBN model with $N$ neurons, each corresponding to one of the $N$ bandit arms;

(2) pick network parameters that yield "optimistic" exploration for a 2-D BBN, and simply apply the parameters to all neurons in the N-D model;

(3) at each trial, sample input $\boldsymbol{I}$ from the reward memory buffer and numerical simulation of the network for $T$ steps using the Runge-Kutta method;

(4) at the end of the simulation, select the arm $a$ whose corresponding neuron has the highest activation value;

(5) collect the reward $r_a$ and add it to memory buffer for arm $a$;

(6) repeat (3)-(5) for the next trial till game ends.

## B.3 Running BBN in MDP tasks

Here we consider the tabular case MDP, so the states and Q-values are parameterized as entries in a lookup table, where each state-action pair maps to a Q-value. To implement BBN in action selection, the agent needs to estimate the uncertainty of Q-values for BBN's input. However, how to estimate uncertainty of the cumulative rewards in MDP tasks remains an open issue in the RL community because the choice of an action affects both the current immediate reward and subsequent state transfer. O'Donoghue et al. (2018) gave an upper bound on the variance of posterior distribution of the Q-values by proposing the Uncertainty Bellman Equation (UBE), which connects the uncertainty at any time-step to the expected uncertainties at subsequent time-steps. We leverage the upper bound on the variance by UBE to obtain uncertainty estimation for Q-values.

Here we present detailed steps to apply BBN to drive action selection in MDP tasks: (1) define a BBN model with $N$ neurons, each corresponding to one of the $N$ discrete actions, select network parameters that belong to the "optimistic" regime for a 2D network;

(2) initialize state-action values to i.i.d. Gaussian distributions;

(3) sample input values for each neuron from the distributions of state-action values and perform numerical simulation of the BBN network for $T$ steps using the Runge-Kutta method;

(4) at the end of the simulation, select action $a$ whose corresponding neuron has the highest activation value;

(5) collect the reward $r_a$ and move to the next state ;

(6) Repeat (3)-(5) till the episode ends;

(7) Update the distribution of state-action values using the uncertainty bellman equation (UBE) algorithm(O'Donoghue et al., 2018).

(8) repeat (3)-(7) for next episode till game ends. We present the pseudo-code for the Algorithm 2 in Appendix B.4. The pseudocode along with detailed task parameters are presented below.

## B.4 Pseudocodes

Algorithm 1 presents the pseudocode of BBN in Multi-armed Bandit Games and Algorithm 2 presents the pseudocode of UBE-BBN in playing MDP tasks.

---

**Algorithm 1:** BBN for multi-armed bandit games

---

**Input** :

        The horizon of the multi-armed bandit game $H$;

        The number of arms $A$ ;

        The total simulation steps for BBN model $T$;

**Output:**

        The selected arm $a$ at each trial $h$;

Initialize the model parameter for BBN model;

Initialize the value for each neuron $x_i$;

**for** $h = 1, 2, ..., H$ **do**

    **for** $t = 1, 2, ..., T$ **do**

        sample $I_i$ from reward history for each arm $a_i$

$$\tau_i \frac{dx_i}{dt} \leftarrow -\gamma_i x_i + \sum_{j \neq i}^{N} w_{ij} f(x_j) + b_i + I_i \ ;$$

        $x_i \leftarrow x_i + dx_i$ ;

    **end**

    select an arm $a \leftarrow argmax(x_i)$;

    receive a reward $r_a \sim \mathcal{N}(\mu_a, \sigma_a^2)$;

    add $r_a$ to reward history of arm $a$;

**end**

---

---

**Algorithm 2:** UBE-BBN for MDP tasks

---

**Input :**

        The horizon of the MDP task $H$;

        The maximum episode $\tau$ ;

        The number of total states $S$ ;

        The number of actions $A$ ;

        The total simulation steps for BBN model $T$;

**Output:**

        The selected action $a$ at each timestep $t$;

Initialize the model parameter for BBN model;
Initialize the value for each neuron $x_i$;
**for** *iter = 1, 2, ..., $\tau$* **do**
    **for** *h = 1, 2, ..., H* **do**
        s $\leftarrow$ current state ;
        **for** *t = 1, 2, ..., T* **do**
            sample $I_i$ from $Q_{si} \sim \mathcal{N}(\hat{Q}_{si}, \mathbf{var}\hat{Q}_{si})$ ;
            $\tau_i \frac{dx_i}{dt} \leftarrow -\gamma_i x_i + \sum\limits_{j \neq i}^{N} w_{ij} f(x_j) + b_i + I_i$ ;
            $x_i \leftarrow x_i + dx_i$ ;
        **end**
        select an action $a \leftarrow argmax(x_i)$;
        receive a reward $r_{sa} \sim \mathcal{N}(\mu_{sa}, \sigma_{sa}^2)$;
        move to next state $s'$ ;
        update $\hat{P}_{sas'}$;
    **end**
    update Q values using dynamic programming:
    **for** *h = H, H-1, ..., 1* **do**
        **for** *$s \in S$* **do**
            **for** *$a \in A$* **do**
                $\hat{\mu}_{sa} \leftarrow \mathbf{E}r_{sa}$;
                $\hat{Q}_{sa}^h \leftarrow \hat{\mu}_{sa} + \sum_{s'a'} \pi_{s'a'} \hat{P}_{sas'} Q_{s'a'}^{h+1}$ ;
                employ the Uncertainty Bellman Equation (UBE):
                $\mathbf{var}\hat{Q}_{sa}^h \leftarrow \mathbf{var}\hat{\mu}_{sa} + \sum_{s'a'} \pi_{s'a'} P_{sas'} \mathbf{var}\hat{Q}_{s'a'}^{h+1}$ ;
            **end**
        **end**
    **end**
**end**

---

### B.5 BANDIT AND MDP TASK PARAMETERS

**Bandit parameters for performance comparison** We chose to use Gaussian bandits where reward values are sampled from $\mathcal{N}(\mu_i, \sigma_i^2)$. For 2-armed bandit games, the reward mean $\mu$ for both arms are sampled from a Gaussian distribution $\mathcal{N}(0, 1^2)$ at the beginning of each block. The reward variance is 9 and 4 respectively. For 3-armed bandit games, the reward mean $\mu$ for all arms are sampled from a Gaussian distribution $\mathcal{N}(0, 1^2)$ at the beginning of each block. The reward variance are 9,1,0.25 respectively. Note that while we chose to use Gaussian bandits here, the model can be extended to non-Gaussian input distributions and performs well empirically in non-Gaussian (e.g. Bernoulli) bandit tasks.

**Bandit parameters for fitting to mice data** We follow the bandit parameters in (Beron et al., 2022). The mean rewards of the Bernoulli bandits are 0.8 and 0.2 respectively.

**SixArms** SixArms(Strehl & Littman, 2008) consists of seven states and six actions. The agent starts in state 0. We consider episodic case, so the state is reset every 20 steps. A transition is of

the form $(a, p, r)$, where $a$ is action, $p$ is the transition probability, and $r$ is the reward for taking the transition.

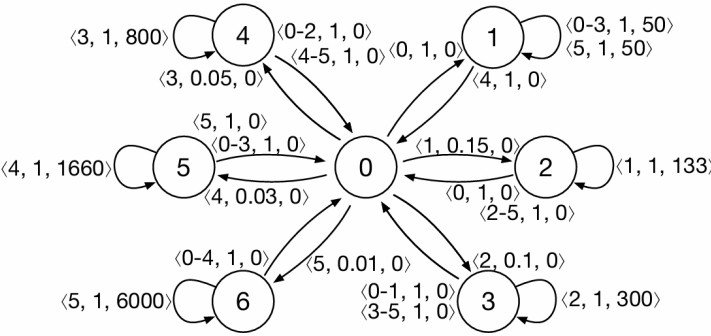

Figure 16: **SixArms.**

For more detailed parameters for each algorithm used in our experiments, please refer to our code: https://github.com/anonymousforICLR/BrainBandit

## C DATASETS FOR MODEL FITTING

**Gershman19** is from (Gershman, 2019). In their experiment, participants were given a choice between two arms, labeled either as "safe" (S) or "risky" (R). The safe arms always return deterministic rewards, while the risky arms sample rewards from a Gaussian distribution. There are four types of bandit settings: RS, SR, RR, and SS, which are denoted by compound labels (e.g., "SR" denotes trials in which the left arm is safe and the right arm is risky). The reward mean $\mu$ for both risky arms and safe arms are sampled from a Gaussian distribution $\mathcal{N}(0, 10^2)$ at the beginning of each block. The reward variance for risky arms is 16, and for safe arm is 0. By comparing the slope and intercept of the choice probability curve for each type, we can quantify the degree of randomness and preference for uncertainty.

**Fan23**(Fan et al., 2023) further explored the relationship between trait somatic anxiety and different exploration strategies in decision-making. They used the same experimental design as **Gershman19**(Gershman, 2019) and evaluated the anxiety for each individual. In Fig 6 (a-b), the slope and intercept of human data in (Gershman, 2019) are drawn directly from the paper. And for humans with high or low anxiety, we split the 40% of the population with the highest "somatic anxiety" score and the 40% with the lowest "somatic anxiety" score in the collected data from (Fan et al., 2023), and then performed probit regression respectively.

**Mizell24** from (Mizell et al., 2024) involved younger adults (ages 18–25) and older adults (ages 65–74) making decisions between two virtual slot machines to measure exploration behaviors called Horizon Task. The rewards are sampled from a Gaussian distribution. Participants first completed instructed trials, sampling the slot machines under two conditions: unequal information (one drawn from one machine and three from the other) and equal information (two drawn from each machine). They then made free choices in either a short horizon (one choice) or a long horizon (six choices) condition. The task assessed directed exploration (choosing the more informative option) and random exploration (choosing the lower reward option). We use unequal information condition of the collected data to fit our model.

**Zajkowsk17** is from (Zajkowski et al., 2017). Participants also performed a Horizon Task, where they made explore-exploit decisions between two virtual slot machines under two conditions: unequal information and equal information. The task involved 160 games, each consisting of 5 or 10 choices, with the key manipulation being the horizon length: short (5 choices) or long (10 choices). Continuous theta-burst transcranial magnetic stimulation (TMS) was used to selectively inhibit the right frontopolar cortex (RFPC) when participants performed the Horizon Task. We use unequal information condition of the collected data to fit our model.

# D    FURTHER RESULTS ON BANDIT TASKS

## D.1    LIMITED MEMORY BUFFER SIZE

BBN doesn't need all the past experience in the memory buffer. For example, in the experiment of fitting to mice behavior, we only used the last 5 reward histories since the reward for each bandit will change over time. We also performed additional experiments to test if the limited memory buffer would hurt the performance in bandit tasks. We limited the buffer size to 8 for each arm. Fig 17 shows BBN with limited memory buffer size still consistently outperforms other methods.

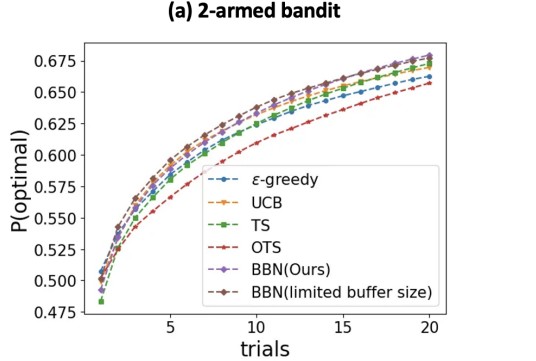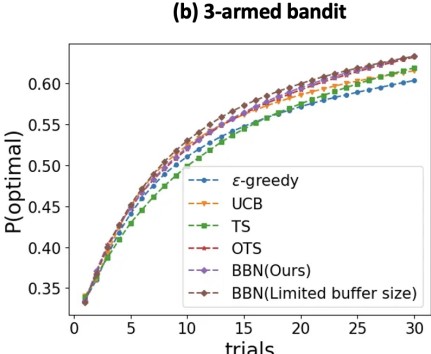

Figure 17: **BBN with limited memory buffer size achieve similar efficient exploration in bandit tasks**

# E    FURTHER RESULTS ON MDP TASKS

## E.1    PARAMETER SENSITIVITY ANALYSIS

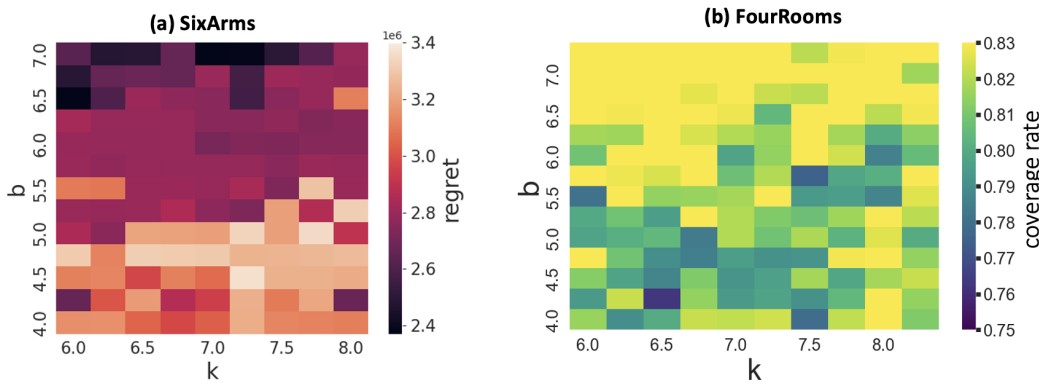

Figure 18: **Parameter sensitivity analysis of UBE-BBN with different parameter combinations evaluated in two MDP tasks.** Performance in the SixArms task was evaluated by the cumulated regret of the agent, while performance in the FourRooms grid world task was evaluated by the coverage rate. a broad range of "optimistic network parameters" generally yielded high performance on these tasks.

## E.2 PERFORMANCE ON VARIATIONS OF GRID WORLD TASKS

As shown in Fig.19, UBE-BBN yields fastest coverage rate among all the methods on different environments. Fig.20 gives examples of cumulative visitation counts for more algorithms during training. Only UBE-BBN covers all states with less than 450 episodes.

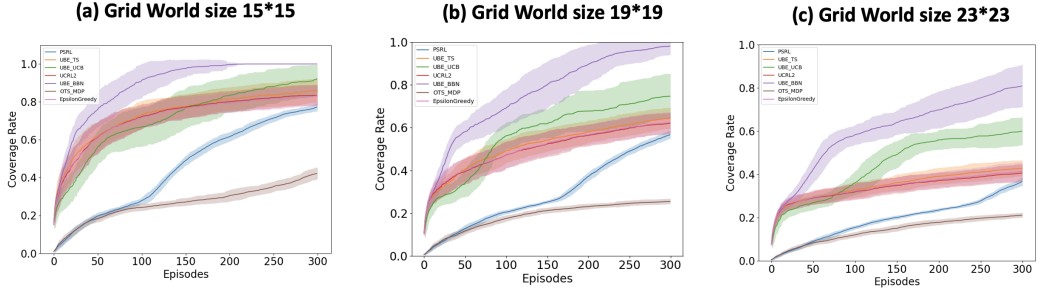

Figure 19: **Learning curves on different sizes of FourRooms environments.**

Fig.24 shows the trajectories of agents, which are the visitation counts in a single episode. As shown, $\epsilon$-greedy, UCRL2, UBE-TS only perform exploration around the starting state, failing to do "deep" exploration. PSRL, OTS-MDP and UBE-UCB can perform "deep" exploration, but they all act deterministically, so they will be stuck at a certain state. UBE-BBN is also driven by uncertainty like UBE-UCB to perform "deep" exploration, but with stochastic sampling of action choices, it will not be stuck at a certain state.

**Action persistence further boosts BBN performance.** BBN with persistence refers to taking neuron values at the end of last step as the starting point for the next step, while BBN without persistence refers to initializing neuron values at each step. We compare the different behavior of the BBN model with and without persistence across four different grid sizes: 15×15, 19×19, 23×23, and 103×103. The results presented here show the trajectories during the first episode of exploration, and the exploration length corresponds to the number of states in the grid world. As shown in Fig. 25, for the same exploration length, the BBN model with persistence explores a larger portion of the grid world.

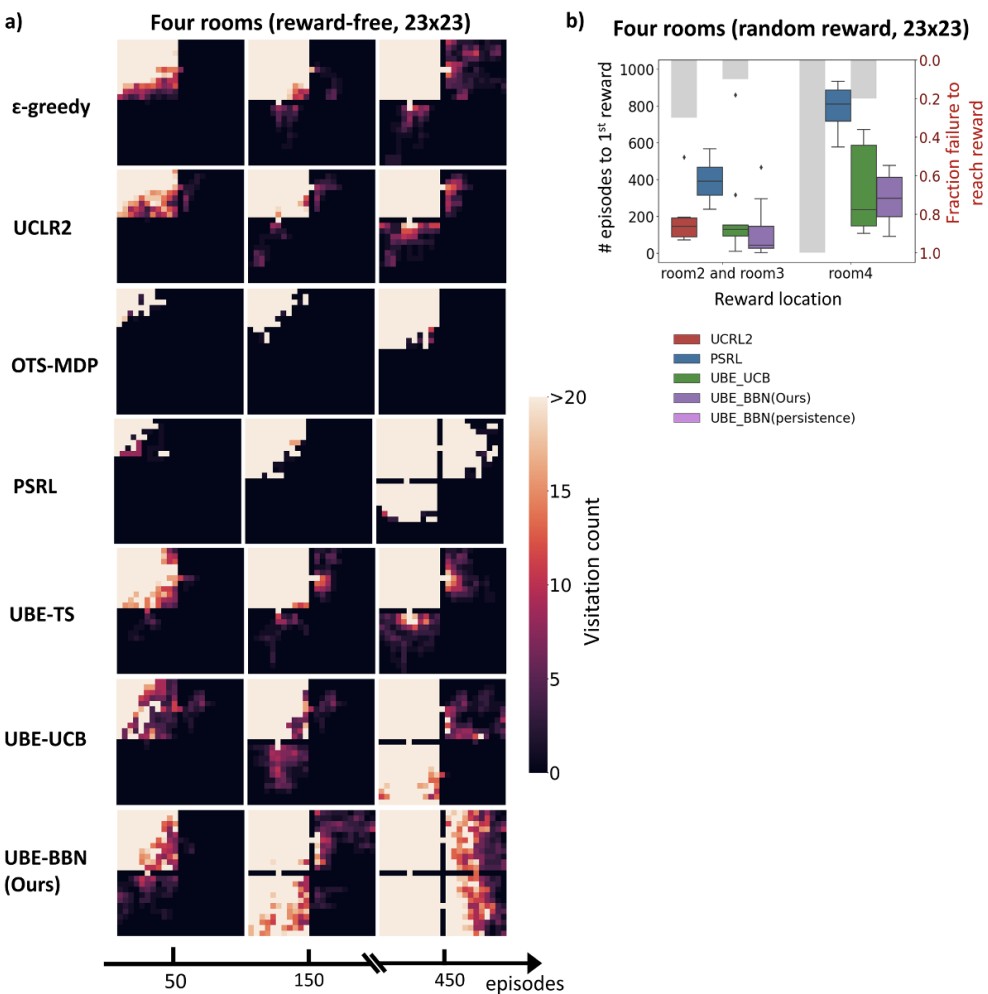

Figure 20: **Comparison of exploration efficiency across different exploration algorithms in Four Rooms)** (a) Visitation counts in reward free setting; (b) Number of episodes until first encounter of the reward state.

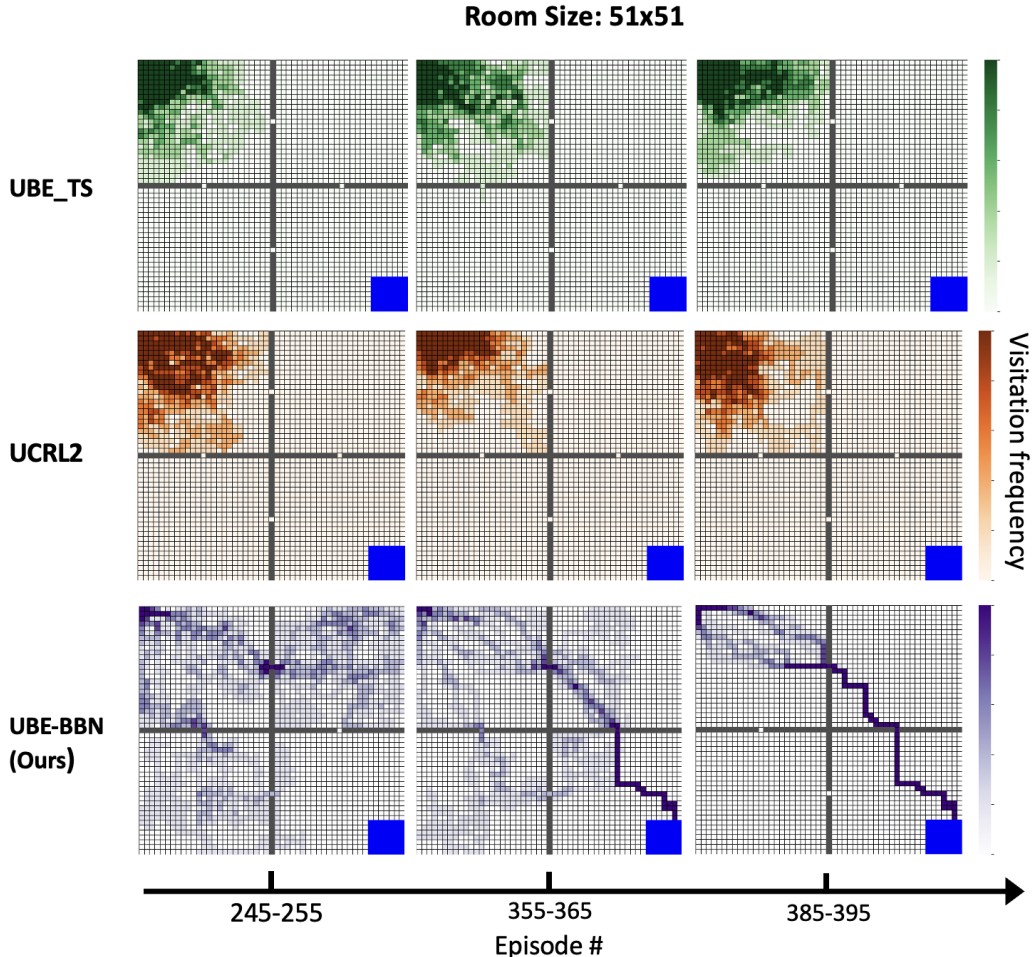

Figure 21: **Trajectories (visitation counts in a single episode) of UBE-TS, UCRL2, and UBE-BBN in expanded Four Rooms task with reward**

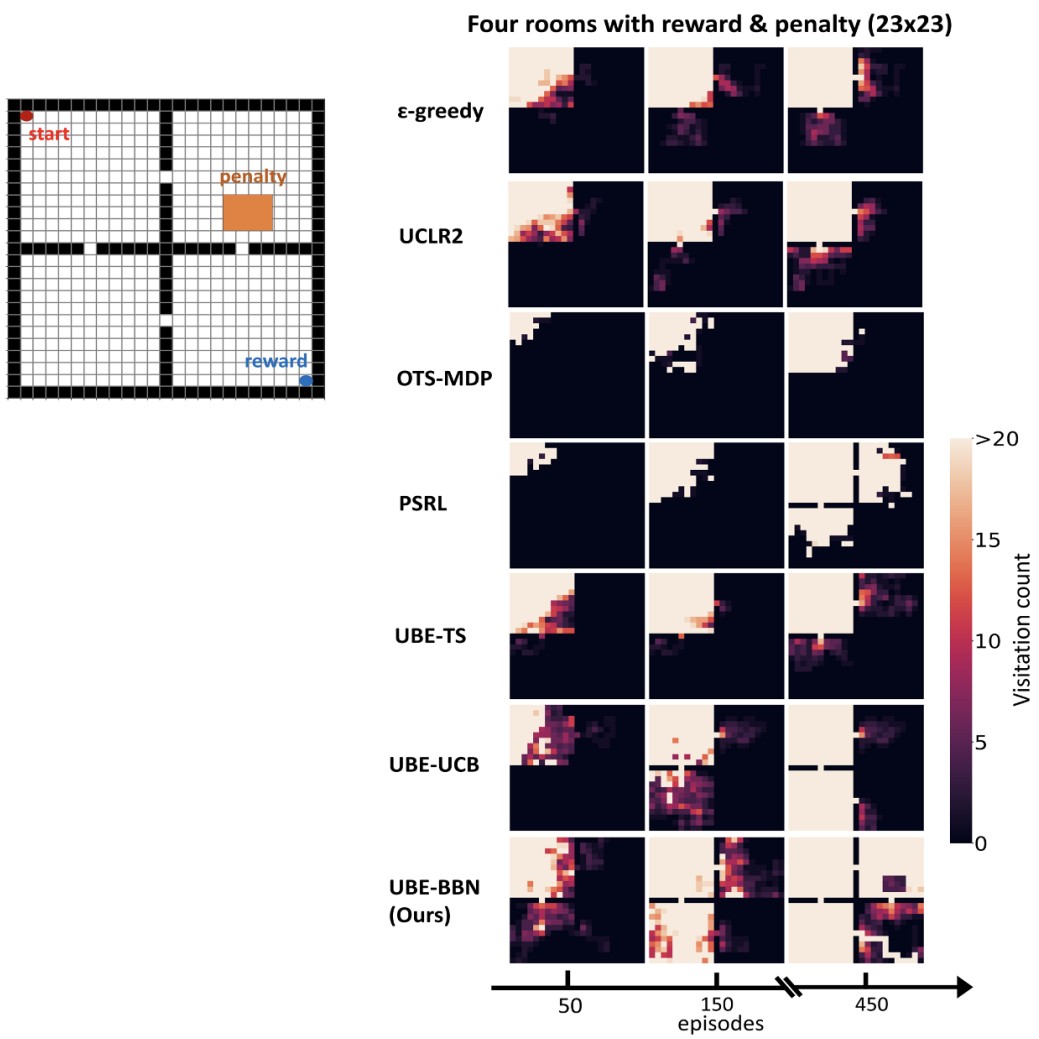

Figure 22:   **Comparison of visitation counts across algorithms in a Four Rooms game with reward and penalty).**

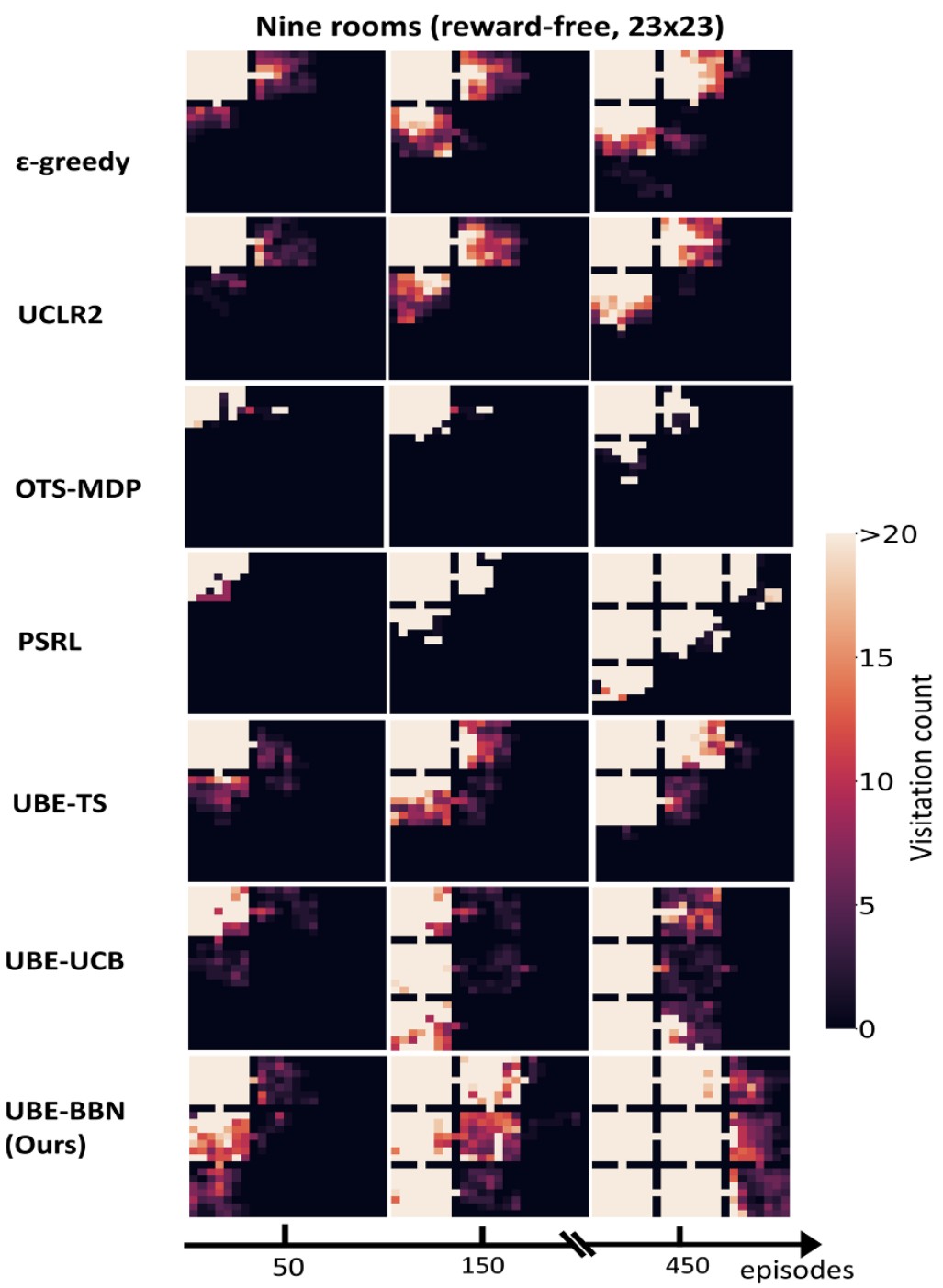

Figure 23: **Comparison of visitation counts across algorithms in a Nine Rooms game.**

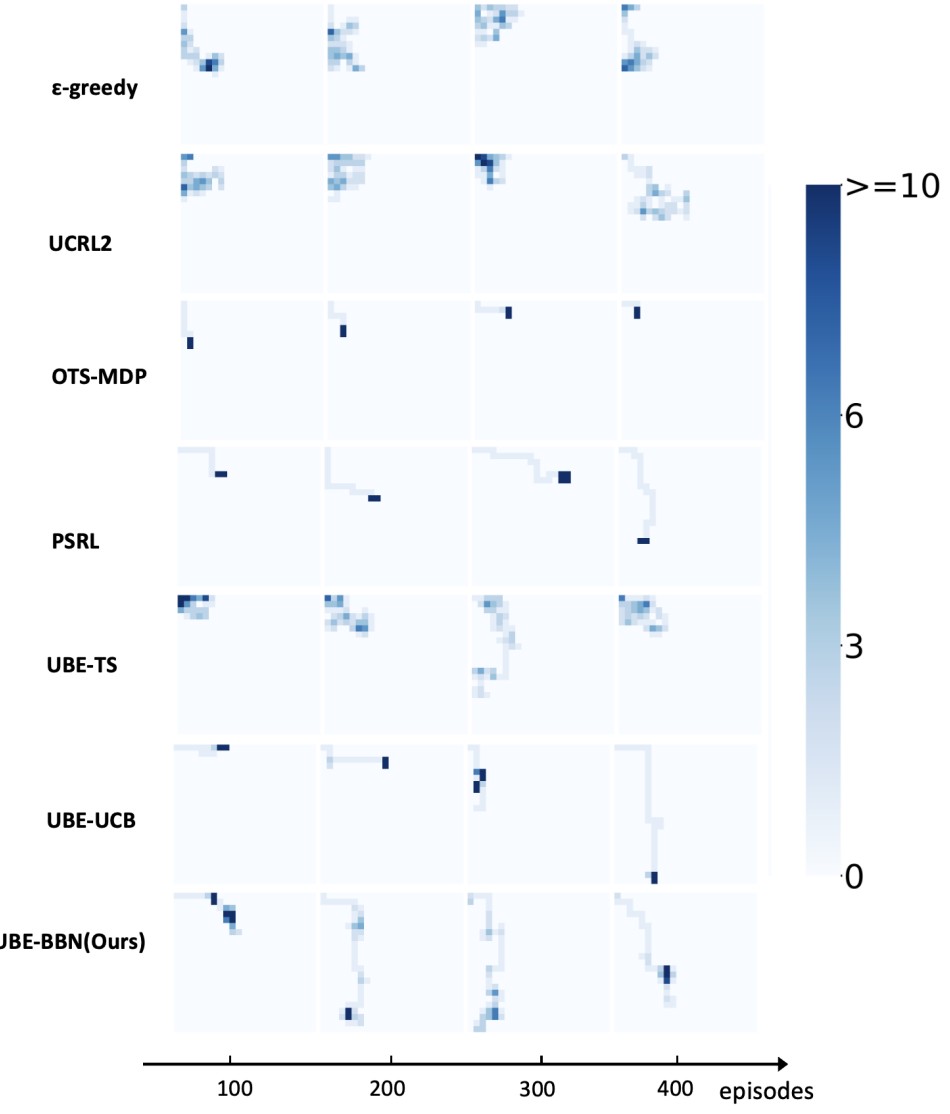

Figure 24: **Agent trajectories (visualized through visitation counts) in single episodes over the course of training.**

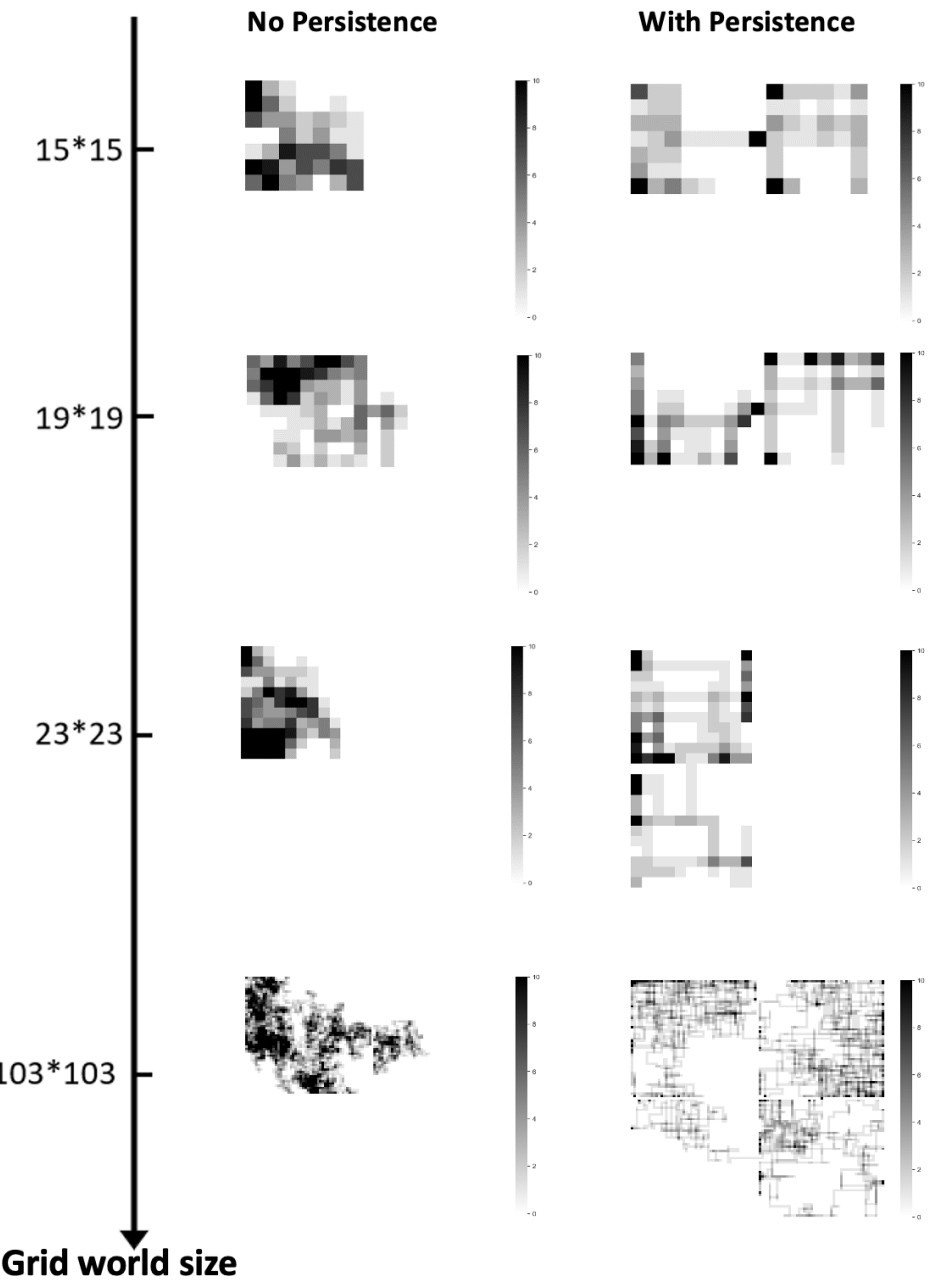

Figure 25: **Trajectories (visitation counts in a single episode) of BBN with/without persistence.**

