# OpenReview forum: "Brain Bandit: A Biologically Grounded Neural Network for Efficient Control of Exploration"
_ICLR.cc/2025/Conference — ICLR 2025 Oral_

### Official Review · Reviewer_8itA · 2024-10-25

**Soundness:** 3
**Presentation:** 3
**Contribution:** 3
**Rating:** 8
**Confidence:** 4

**Summary:**

In this work, the Authors propose a framework addressing the exploration-exploitation tradeoff using a stochastic continuous Hopfield network. They provide a derivation to show that the proposed framework implements a form of Bayesian inference while also displaying an uncertainty-related bias. The comparison of the proposed model with human and mouse data on multi-armed bandit task shows the similarity between the two; additional considerations and experiments are provided for two MDP tasks.

**Strengths:**

The work deals with an interesting and important problem of the exploration-exploitation tradeoff, particularly focusing on the aspect of human decision-making. As human decision-making, shaped by evolution, is expected to be optimal, at least in certain ways, understanding its mechanics may be useful for the design of future ML algorithms. This work combines the study of exploration-exploitation tradeoff with recurrent neural networks and bandit tasks, which are all relevant and interesting to the ICLR community.

The quality of the research in this work is high: it starts with the thorough derivations of the analytic predictions for the system’s behavior and then continues with numerical experiments where the model’s predictions are first validated and then comparisons are performed with human and animal data. The text is mostly well-written, following the standard structure of machine-learning papers and introducing concepts in a logical sequence, making it easy to follow.

The result regarding the similarity between the model’s predictions and human/mouse data on exploration/exploitation in multi-armed bandit tasks is novel and interesting. It is particularly strong as it spans multiple datasets from multiple works.

**Weaknesses:**

While the derivations for the analytic results are thorough and lengthy, most of them appear to be either existing or intuitive. The attractor structure of the Hopfield network with two neurons and inhibitory connections is known. Hopfield networks have been shown to be able to implement Bayesian inference, although the derivations have been provided for the discrete case. The fact that the probability of the network’s activity to stay in an attractor depends on the relative orientation of its Hessian matrix and the noise direction is clear and intuitive. Of course, none of that diminishes the novelty of the result regarding the similarity between the model’s behavior and human decision-making.

The MDP part may need a fuller description (ideally a Methods section; e.g. in the Appendix) to clarify what gives rise to the observed phenomena. The uncertainty Bellman equation is also worth mentioning in the text for the clarity. Please see the specific questions regarding the Gridworld task below.

**Questions:**

How were the states and Q-values parameterized in the task? Please provide a Methods-style description of the model. This may be particularly important in the light of the next question.

One thing that I couldn’t understand is: What makes the model explore more than the baselines? In particular, how would the exploration in this task compare to the random walk? How would that result change based on the parameterization of the stare and the Q-function (e.g., tabular vs. deep learning)?

Finally, I didn’t quite understand why is the higher state coverage considered a positive thing in this task. Specifically, as the task is not rewarded (Figure 8 b-c), the exploration seems unnecessary. When rewards are present, their utility may be different depending on the reward and the task.

Overall, I think this is an interesting paper relevant to the ICLR community. I believe that it offers sufficient novelty and significance through the remarkable match between the model’s predictions and human behavior. Text-related issues can be addressed during the remaining time.

________________
Post-rebuttal: questions addressed by the Authors; raising the score to 8.

---

> ### Author Response · Authors · 2024-11-22
>
> We deeply appreciate the constructive feedback provided by the reviewer. We also thank the reviewer for bearing with us as we revise our manuscript. In response to the potential concerns and questions:
>
> 1. Regarding the novelty of our analytical results: While Bayesian inference in discrete Hopfield networks and the role of flat minima in stochastic gradient descent have been studied before, our theoretical analysis makes novel contributions by combining these frameworks to explain how common brain circuits (specifically reciprocal inhibition) arbitrate between exploration and exploitation. Although attractor networks have previously been used to model probabilistic decision-making, we are among the first to theoretically show that Hopfield networks can flexibly modulate their bias toward or against uncertainty in ways that match human and animal behavior. Our work thus advances our understanding of how the brain balances exploration and exploitation, while offering insights that could improve machine learning algorithms.
> 2. More complete description on the application of BBN to MDP problems:  Thanks to this suggestion, we have now added more detailed step-by-step descriptions on how to set up a BBN model and embed it in a MDP problem in Appendix B.1 & B.2.1 of the revised article.

---

> ### Author Response · Authors · 2024-11-22
> **Answers to specific questions**
>
> Answers to specific questions regarding BBN performance in the grid world task:
>
> (1) How are states and Q values parameterized (need method style description)?
>
> We implemented the grid world environment as a tabular MDP, storing states and Q-values in a lookup table where each state-action pair maps to a Q-value. To estimate Q-value uncertainty, we used the Uncertainty Bellman Equation (UBE, O'Donoghue et al. 2018), which effectively connects local reward uncertainty to future expected uncertainties in tabular MDP problems. We specifically used the upper bound on Q-value variance computed through UBE as our variance measure. Following the step-by-step method detailed in Appendix B.1 and B.2.1, we set up the 4D-BBN model and ran it in various grid world tasks to generate action choices (Fig. 7 and 20-23).
>
> (2) What makes the model explore more than baselines? How would the results depend on the tabular parameterization of state and Q values?
>
> As our theoretical analysis shows, the BBN model combines two powerful exploration strategies: stochastic posterior sampling and optimism in the face of uncertainty. In sparse reward tasks like SixArms or Four Rooms, the agent must efficiently explore a large state space to have its first encounter with a reward. Pure random walk (equivalent to epsilon-greedy in the reward-free setting) expands its exploration frontier slowly, as shown in Fig. 7 and additional experiments in Appendix E.2, because it does not bias exploration toward unvisited states. UCB-type algorithms (e.g., UCRL2, UBE-UCB) direct exploration toward less visited states but do so deterministically, which can lead to biased uncertainty estimates. Meanwhile, posterior sampling or Bayesian sampling requires an efficient sampler—BBN, formulated as Langevin SDEs, implements Langevin sampling, which is known for its high efficiency. We believe these combined attributes make BBN particularly effective at performing deep and wide exploration of the task environment.
>
> The BBN model is technically compatible with both linear function and neural network approximation of Q-values, provided there exists a reliable estimator for cumulative reward uncertainty. However, the UBE method loses its theoretical guarantees when extended to function approximation and has performed poorly in our empirical tests. Simple approaches—such as using local reward uncertainties estimated through Bayesian linear regression in the final network layer—prove inadequate for tasks like Four Rooms that demand deep exploration. Despite testing various global reward uncertainty estimators, we have not yet found one that can accurately estimate global uncertainty while remaining compatible with function approximation. We aim to address this challenge in future work, possibly by studying how brain circuits estimate future uncertainty.
>
> (3) Why is high state coverage beneficial for grid world tasks? Does the benefit of exploration depend on specific designs of reward and task?
>
> High state coverage is particularly important for sparse reward tasks that require deep exploration (e.g. the Four Rooms task with one single reward located in the farthest corner, as illustrated in Fig. 7a). This is because initially, the agent is unaware of the reward's location, necessitating exploration of as much grid space as possible. Once the agent has the first encounter with the reward location, it can rapidly learn the optimal trajectory. High state coverage is particularly challenging in Four Rooms because the chance of passing through the narrow passage between rooms is low. As illustrated in Fig. 21 of Appendix E.2, the UBE-BBN agent has reached the reward by episode 355, after which it quickly learned the optimal trajectory to the reward location. In contrast, the UBE-TS and UCRL2 agents have yet to enter the room with reward at this stage.
>
> We believe the benefit of exploration does not depend sensitively on specific designs of the reward and task. To support this, we have evaluated 4D BBN performance in multiple variants of the grid world tasks (Figs. 20-23 in Appendix E.2). These include Four Rooms tasks with fixed (Fig 7d&e) and randomly sampled reward locations (Fig. 20), Four Rooms with larger grid size (Fig 7d & Fig. 21), Four Rooms with both reward and punishment (Fig. 22), and a 9-room grid world requiring deeper exploration (Fig. 23). In all scenarios, optimistic BBN surpassed other methods and most efficiently learned optimal reward paths. While we have not tested all possible grid world task designs, our initial results do confirm BBN's robust performance across task settings.
>
> Once again, we wish to thank the reviewer for the insightful comments which helped us improve our paper.

---

> > ### Comment · Reviewer_8itA · 2024-11-25
> >
> > Thank you for clarifications. They address my questions; I'm happy to raise my score.

---

> > > ### Author Response · Authors · 2024-11-27
> > >
> > > Thank you for carefully reading our revised paper and updating your assessment. We greatly appreciate your constructive comments and clarifying questions, which helped improve the paper. We will continue working on the important issues you raised (particularly regarding effective application to deep learning). Please stay tuned for future developments. Thank you again!

---

### Official Review · Reviewer_aJhq · 2024-11-04

**Soundness:** 3
**Presentation:** 2
**Contribution:** 3
**Rating:** 8
**Confidence:** 3

**Summary:**

# Summary
The authors present a biologically-inspired neural network model called Brain Bandit Network (BBN) that addresses the explore-exploit problem in sequential decision making tasks.
The model itself is a stochastic continuous Hopfield network, which they show can implement a form of Bayesian posterior sampling and a tunable input uncertainty bias.
They demonstrate the BBN's effectiveness in MAB tasks, showing that it approximates human and animal choice patterns, and on some MDP exploration benchmarks.

**Strengths:**

# Strengths
* Novelty: As far as I can tell, this is a novel combination of Stochastic Hopfield Networks applied to decision making / explore-exploit tasks. The connections with neuroscience are interesting.
* Well written/Thorough: Paper is quite clear and well written, and as such has high pedagogical value. The authors present extensive experiments comparing BBN against baseline methods on multiple tasks, providing empirical support for their claims. (Minor concerns addressed below)

**Weaknesses:**

# Weaknesses
* Niche/limited applicability: The method itself is quite niche, needs an SDE solver, and has several hyperparameters that require niche/expert knowledge to tune.
* Tuning the numerous hyperparameters requires specialized knowledge, making it difficult for non-experts to utilize effectively. Are there principled ways to determine optimal BBN parameters for new environments?
* The performance of this model in high-dimensions needs more exposition. Section 3.4 provides a start, but more extensive experiments and theoretical analysis are needed to understand the behavior and performance of BBN in high-dimensional settings. (ideally supported by some theoretical footing).

**Questions:**

# Suggestions for improvement:
* L169 -- L174: What is MFPT? The lead up to these equations needs better development, esp. for someone not already familiar with this literature
* L202/208: Define escape efficiency
* L322/323: Define relative and total uncertainty
* L438 and elsewhere: Explain the rationale behind the choice of T steps for BBN simulation.
* Citation format seems wrong

---

> ### Author Response · Authors · 2024-11-22
>
> We sincerely thank the reviewer for recognizing the novelty and quality of our study. We are also truly grateful for the constructive feedback. Regarding the concerns raised by the reviewer:
>
> (1). Niche/Limited applicability: The reviewer raised valid points about how using an SDE solver and fine-tuning multiple parameters could limit general applicability. However, for BBN, we found that simple Euler discretization with a fixed step size was sufficient. Using our suggested parameter ranges (see Tables 1-3 in Appendix B.1), the model could transition between attractor states multiple times within a few hundred simulation steps (a desired property for posterior sampling), and outperformed UCB and TS-based algorithms in exploration efficiency. Furthermore, our parameter sensitivity analysis on MDP tasks (Fig. 18 of Appendix E.1) showed that a wide range of "optimistic" network parameters produced high empirical performance. This demonstrates that extensive fine-tuning is not necessary to achieve an optimistic bias that enhances exploration.
>
> (2). Regarding hyperparameter tuning: In the newly added Appendix B.1, we present the full list of parameters for setting up a BBN model. These include internal model parameters (which specify neuronal dynamics and inter-neural connections), input normalization parameters, and numerical simulation hyperparameters. For all 11 parameters, we provide suggested value ranges and explain how each affects model dynamics. These ranges have proven effective across all experiments in our paper. Fine-tuning the bias is straightforward—it can be achieved by adjusting either \(b\) or \(k\), as shown in our parameter sensitivity analyses (Fig. 3(a-b)). We also provide step-by-step instructions for setting up an \(N\)-dimensional BBN with optimistic bias for both bandit and MDP tasks in Appendix B.1-3.
>
> (3). Regarding the analysis of BBN performance in high dimensions with theoretical support: Following the reviewer's suggestion, we conducted a more comprehensive analysis of BBN performance in high dimensions. We first examined the agreement between theoretical and numerical state probabilities in high-dimensional BBNs up to N=10 (Fig. 12). As noted in section 3.4, the network showed an increasing bias toward high-uncertainty states as network dimension increased. This effect appears in both theoretically neutral and conservative networks beyond N=5.
>
> To understand the mechanism behind this phenomenon, we analyzed state-transition dynamics near the saddle point for a perfectly neutral 3D BBN (Fig. 13). With isotropic noise, the network showed equal probability of entering any attractor state. However, highly anisotropic noise caused the network to preferentially enter attractor states along the dimension of the highest noise, creating a bias toward high-uncertainty states. This noise anisotropy affects the rate of entering (rather than leaving) attractor states, making it particularly difficult to maintain a conservative bias in high-dimensional models (as demonstrated in Fig. 14, where a 4D network shows greater deviation from theoretical predictions). Integrating this effect into our theoretical framework would require combining escape rates analysis with saddle point dynamics theory—a challenge for future research.
>
> Finally, beyond our evaluations of 3D, 4D, and 6D BBN in bandit and MDP tasks, we tested 4D BBN performance across multiple grid world variants (Figs. 20-23 in Appendix E.2). These include Four Rooms tasks with fixed (Fig 7d&e) and randomly sampled reward locations (Fig. 20), Four Rooms with larger grid size (Fig 7d & Fig. 21), Four Rooms with both reward and punishment (Fig. 22), and a 9-room grid world requiring deeper exploration (Fig. 23). In all scenarios, optimistic BBN surpassed other methods and most efficiently learned optimal reward paths. These results confirm BBN's robust performance across various tasks and network dimensions.

---

> ### Author Response · Authors · 2024-11-22
> **Answers to specific questions**
>
> Answers to specific questions:
>
> 1. Define MFPT and provide background on the related literature: According to thermodynamics, the mean first passage time (MFPT) is defined as **the average time it takes for a diffusing particle to reach a target position for the first time**. Because the Langevin or SDE formulation of the BBN is equivalent as describing the dynamics of the model as a drift-diffusion process, we naturally employ the concept of MFPT to define the expected time it takes for the model to escape from the center of an attractor and reach the nearest saddle point. We apologize for missing to define the MFPT concept in the previous manuscript and have now included the definition in line 148-150.
> 2. Define escape efficiency: Escape efficiency is defined as the expected rate at a dynamical system leaves one of its attractor states, which is also the inverse of the mean first passage time. We have now added the definition in line 197-198.
> 3. Define relative and total uncertainty: Following the convention in the human bandit task literature (Gershman et al. 2018&2019), we define the relative uncertainty as $\sigma_1- \sigma_2$, and the total uncertainty as $\sqrt{\sigma_1^2+\sigma_2^2}$, where $\sigma_1$ and $\sigma_2$ are standard deviation of external input. We have now included the mathematical definitions in line 230-232.
> 4. Explain the choice of T, the number of simulation steps for BBN: The value of T reflects a balance between simulation accuracy and computational efficiency. A longer simulation time enables more thorough sampling of attractor states, leading to more precise estimates of equilibrium state probabilities. For detailed analysis of model behavior, we used T > 100,000 to maximize precision. However, for practical applications in bandit and MDP tasks, we found T = 400 strikes an optimal balance between accuracy and speed. We provide a detailed discussion of T selection in Appendix B and demonstrate example network dynamics for a 2-armed bandit game using T = 400 (Fig. 15).
> 5. Correct the citation format: Thanks for pointing this out. We have now corrected the citation format in the revised manuscript.
>
> Once again, we sincerely thank the reviewer for the valuable feedback, which greatly helped us improve our paper. Also, thank you for bearing with us as we revise our manuscript.

---

> > ### Comment · Reviewer_aJhq · 2024-11-24
> >
> > Thank you for addressing my comments. The clarifications and material added to the appendix should make the paper more accessible to readers.

---

> > > ### Author Response · Authors · 2024-11-27
> > >
> > > Thank you so much for the positive evaluation and all the constructive suggestions. While we haven't yet resolved all the important issues you pointed out (particularly regarding theory and scalability), we are committed to addressing them in our future work. Thank you again!

---

### Official Review · Reviewer_siYM · 2024-11-04

**Soundness:** 3
**Presentation:** 3
**Contribution:** 3
**Rating:** 8
**Confidence:** 3

**Summary:**

This paper presents a simple recurrent neural network model of exploratory action selection.

**Strengths:**

This paper extends the concept of Boltzmann machine to continuous valued recurrent neural networks. I could not follow all the details of the mathematics, but the analysis seems to be new and helpful.
The simulation results are compared with human behavioral data.

**Weaknesses:**

The implementation of bandit assumes the availability of the memory of the all past experience in the memory buffer, which may be impractical as a cognitive model.

**Questions:**

The pseudo code for the MDP tasks includes P_{sas'}. Does this mean that it is assumed that the agent has access to the true dynamics of the environment?

---

> ### Author Response · Authors · 2024-11-22
>
> We thank the reviewer for the constructive comments and for the clarifying questions.
>
> Regarding the reviewer's concern about using all past reward history for bandit games, we'd like to clarify that BBN doesn't require the entire past experience in its memory buffer. For instance, in our experiment fitting mice behavior, we only used the last 5 reward histories as the reward probabilities are updated probabilistically following a Markov process. We also conducted additional experiments to test whether a limited memory buffer would impact performance in bandit tasks. We restricted the buffer size to 8 for each arm and included a new figure in Fig. 17 of Appendix D.1 of the revised article. Notably, the limited memory buffer did not impact performance and BBN still outperformed other exploration methods.
>
> As for the question about the agent's access to the true dynamics of the environment, our answer is No. During learning, the agent estimates the state transition probabilities from past experience. Therefore, the $P_{sas'} $ in the pseudo-code refers to the estimated probabilities, not the true dynamics. We have added a new section in Appendix B.2.1 to explain step-by-step how UBE-BBN is implemented and have updated Algorithm 2 to clarify this point.
>
> In addition, we have performed more analysis to verify the agreement between theory and numerical simulations when in high dimensional BBNs (Figs. 12-14), and experiments to evaluate the performance of BBN in complex, sparse reward MDP problems (Figs. 20-23). Please refer to our revised manuscript PDF for more information.
>
> Once again, we thank the reviewer again for the insightful comments to help us improve our research. Also, thank you for bearing with us as we revised our manuscript.

---

> > ### Comment · Reviewer_siYM · 2024-11-26
> >
> > Thank you for your clarification and revision. I have upgraded my score.

---

> > > ### Author Response · Authors · 2024-11-27
> > >
> > > Thank you so much for reading through our revised manuscript, and for updating your assessment! We really appreciate your valuable feedback and questions！

---

### Meta-Review · Area_Chair_WNyU · 2024-12-26

**Metareview:**

This paper introduces the brain bandit network (BBN), a biologically-inspired stochastic continuous Hopfield network that addresses the exploration-exploitation dilemma in reinforcement learning. The authors demonstrate that BBN can perform posterior sampling with tunable uncertainty bias, matching human and animal choice patterns in multi-armed bandit (MAB) tasks. The model also shows promise when integrated with RL algorithms for MDP tasks. The key contribution is showing how biological neural circuits (specifically reciprocal inhibition) can effectively arbitrate between exploration and exploitation.

Positive:

- new combination of stochastic Hopfield networks with exploration strategies
- good theoretical foundation backed by thorough mathematical analysis
- good empirical validation against both human/animal behavior and baseline algorithms
- extensive experimental evaluation across multiple tasks and scenarios

Weaknesses:

- complexity due to requiring SDE solver and setting multiple hyperparameters
- limited analysis of scalability to high-dimensional problems
- reliance on tabular MDP formulation, with challenges in extending to function approximation

The authors have adequately addressed reviewers' concerns during the rebuttal period. I recommend acceptance.

**Additional Comments On Reviewer Discussion:**

The rebuttal period focused on three main concerns that were addressed by the authors:

- implementation complexity and hyperparameter tuning - authors provided comprehensive parameter guidelines and sensitivity analyses showing robust performance across parameter ranges
- high-dimensional scaling - authors added extensive analysis of high-dimensional BBN behavior (up to N=10) with theoretical backing
- MDP implementation details: authors included detailed MDP implementation descriptions and additional experiments across various grid-world variants

The authors also clarified questions about memory requirements for bandit tasks and state transition probability estimation. All reviewers found the responses satisfactory.

---

### Decision · Program_Chairs · 2025-01-22

Accept (Oral)